# Small GTPases and BAR domain proteins regulate branched actin polymerisation for clathrin and dynamin-independent endocytosis

Mugdha Sathe[1], Gayatri Muthukrishnan[1], James Rae[2,3], Andrea Disanza[4,5], Mukund Thattai [1,6], Giorgio Scita [4,5], Robert G. Parton [2,3] & Satyajit Mayor[1,7]

Using real-time TIRF microscopy imaging, we identify sites of clathrin and dynamin-independent CLIC/GEEC (CG) endocytic vesicle formation. This allows spatio-temporal localisation of known molecules affecting CG endocytosis; GBF1 (a GEF for ARF1), ARF1 and CDC42 which appear sequentially over 60 s, preceding scission. In an RNAi screen for BAR domain proteins affecting CG endocytosis, IRSp53 and PICK1, known interactors of CDC42 and ARF1, respectively, were selected. Removal of IRSp53, a negative curvature sensing protein, abolishes CG endocytosis. Furthermore, the identification of ARP2/3 complex at CG endocytic sites, maintained in an inactive state reveals a function for PICK1, an ARP2/3 inhibitor. The spatio-temporal sequence of the arrival and disappearance of the molecules suggest a mechanism for a clathrin and dynamin-independent endocytic process. Coincident with the loss of PICK1 by GBF1-activated ARF1, CDC42 recruitment leads to the activation of IRSp53 and the ARP2/3 complex, resulting in a burst of F-actin polymerisation potentially powering scission.

[1] National Centre for Biological Science (TIFR), Bellary Road, Bangalore 560065, India. [2] Institute for Molecular Bioscience, University of Queensland, Brisbane, QLD 4072, Australia. [3] Centre for Microscopy and Microanalysis, University of Queensland, Brisbane, QLD 4072, Australia. [4] IFOM, Fondazione Istituto FIRC di Oncologia Molecolare, Milan 20139, Italy. [5] Department of Oncology and Hemato-Oncology, University of Milan, Milan 20122, Italy. [6] Simons Centre for the Study of Living Machines, National Centre for Biological Sciences (TIFR), Bellary Road, Bangalore 560065, India. [7] Institute for Stem Cell Biology and Regenerative Medicine, Bellary Road, Bangalore 560065, India. These authors contributed equally: Mugdha Sathe, Gayatri Muthukrishnan. Correspondence and requests for materials should be addressed to S.M. (email: mayor@ncbs.res.in)

Multiple endocytic pathways function in a eukaryotic cell[1,2]; however, our understanding of the endocytic process is mainly derived from studies on clathrin-mediated endocytosis (CME)[3–5]. Dynamin is responsible for vesicle scission in CME[6,7] and a host of clathrin-independent endocytic (CIE) pathways, such as the caveolar and fast endophilin-mediated endocytic pathway[8–10]. On the other hand, among CIE pathways, the CLIC/GEEC [clathrin and dynamin-independent carriers which form GPI-enriched endocytic compartments; CG] pathway functions independent of both clathrin and dynamin in multiple cell types and contexts[11–17], while it is not present in others[18]. The actin polymerisation machinery has been implicated in the functioning of many CIE pathways at different stages[13,19].

Our focus, the CG pathway, is regulated by the small GTPases, ARF1 (ADP-ribosylation factor 1) and CDC42 (cell division control protein 42)[11–16]. It is responsible for the uptake of many glycosylphosphotidylinositol (GPI)-anchored proteins, a major fraction of the fluid phase, toxins such as *Helicobacter pylori* vacuolating toxin A[20], cholera toxin[21] and viruses like adeno-associated virus 2[22]. The CLICs are formed in a polarised manner at the leading edge of migrating cells[23] and, the resulting GEECs subsequently fuse with the sorting endocytic vesicles via a Rab5/phosphatidylinositol-3-kinase-dependent mechanism[24]. The CLICs/GEECs are high capacity endocytic carriers turning over the entire membrane surface in 12 min in fibroblasts, highlighting the role of CG pathway in regulating membrane homoeostasis[23]. Recent evidence suggests that this is required for generating a tubular vesicular endocytic network during cytokinesis[25] and serves to deliver ligands to their signalling receptors in a common endocytic compartment[26].

The molecular machinery to form a CG endocytic vesicle involves activating ARF1 at the plasma membrane by GBF1 (Golgi-specific brefeldin A resistance factor 1)[16], a specific ARF-GEF (guanine nucleotide exchange factor). GTP–ARF1 recruits ARHGAP21 (a RhoGAP for CDC42), which removes CDC42 from the membrane[14]. Cholesterol removal, in addition, reduces the recruitment of ARF1 and CDC42, along with accelerated cycling of CDC42[13,14]. Lastly, the CG pathway requires dynamic actin since both stabilisation and de-polymerisation of actin filaments was found to affect endocytosis[13].

By visualising a forming CG endocytic vesicle, we wanted to understand the molecular mechanism responsible. We adapted a pH pulsing protocol that exploits the pH-sensitive properties of super ecliptic GFP (SecGFP)[27], previously deployed to study CME[4,28,29]. We tagged the GPI-anchor with SecGFP to make model cargo SecGFP-GPI to assay, in real time, the sites of endocytic vesicle formation. We found that the CG endocytic vesicle formation was initiated by the accumulation of ARF1/GBF1 followed by CDC42 and F-actin while dynamin and clathrin did not associate with forming endosomes. Hence, in the absence of a discernable coat[23], alternate candidate proteins by generating/stabilising membrane curvature can assist in endocytic vesicle formation such as Bin/Amphiphysin/Rvs (BAR) domain-containing proteins (BDPs)[30].

Although several BDPs are involved in the CME pathway, only one has been identified to be associated with the CG endocytic pathway so far[31]. Using RNAi-screening, we identified two BDPs in particular, that affected CG endocytosis downstream of ARF1 and CDC42. First, a CDC42 interaction partner and I-BAR protein, IRSp53 (Insulin-responsive protein of mass 53 kDa) was found to be necessary for CG endocytosis. Importantly, IRSp53 removal resulted in the disappearance of CLICs and loss of a GBF1-dependent endocytic pathway. Second, an ARF1 interactor, PICK1 (protein interacting with C kinase 1) emerged as a regulator of ARP2/3 activity in the early phases of CG vesicle

formation. Lastly, ARP2/3, an interaction partner of both IRSp53 and PICK1, accumulated at the forming CG endocytic site and decreased CG endocytosis when inhibited. Together, the spatio-temporal dynamics of these proteins provided a mechanistic understanding of the forming CG endocytic vesicle.

## Results

**pH pulsing assay detects nascent CG endocytic sites.** To monitor endocytic vesicle formation in real time, we employed the pH-sensitive fluorescence of super ecliptic pHlourin GFP (SecGFP)[27] attached to a GPI-anchor (SecGFP-GPI) to differentiate cell surface-resident molecules from the newly internalised molecules. The fluorescence of SecGFP is quenched reversibly when exposed to pH 5.5[4,27–29]. SecGFP-GPI expressed in AGS (human gastric cell line) was endocytosed along with the CG cargo, fluid phase (10 kDa dextran), but not with CME cargo, TfR, at both 37 °C and 30 °C (Supplementary Fig.1a–c and Supplementary Information (S.I.)) as shown previously[15,32]. Endocytic events were identified by alternately exposing the cells to buffers equilibrated to pH 7.3 (pH 7) or pH 5.5 (pH 5) every 3 s at 30 °C (Fig. 1a, Schematic and Supplementary Movies 1–2). SecGFP-GPI-containing endocytic events occurring during exposure to pH 7 remained fluorescent due to their near neutral luminal pH right after formation. However, the buffer exchange from pH 7 to pH 5 quenched the fluorescence of cell surface SecGFP-GPI. This enabled visualisation of the newly formed endocytic vesicle. The identification of the site of endocytic vesicle formation paved the way for the characterisation of the spatial and temporal dynamics of molecular players by the co-expression of a (mCherry/TagRFPt/pRuby)-tagged molecule of interest, 'X-FP' (Fig. 1a and Supplementary Movies 1–2). The dynamics of X-FP were extracted by looking at the history of the region, where vesicle formation was detected (Fig. 1a montage (bottom), see Methods and S.I.). To rule out the effect of pH 5 on the rate of endocytosis, we pre-treated the cells with either pH 5 or pH 7 buffer, followed by 5-min pulse in pH 7 buffer and found no difference in CG endocytosis (Supplementary Fig. 1e).

The pH pulsing movies were analysed using semi-automated scripts (see Methods and S.I.). Briefly, the centroid of new spots appearing in the pH 5 channel provided a fiduciary marker for the time and location of the nascent endocytic vesicle (Supplementary Fig. 1d, step 1 and S.I.). The relative enrichment of SecGFP-GPI and X-FP at the endocytic site was determined by normalising the average fluorescence of the nascent endocytic spot to its local background annulus (Supplementary Fig. 1a, d, step 2–3 and see S.I.). The spots were then put through a series of automated and manual checks. The automated check ensured that the pH 5 intensity of the new spot had (i) significantly higher intensity than the background, (ii) persisted for at least 6 s and (iii) did not show an increase in intensity in the subsequent frame. Subsequently, a manual check was performed on the montages, (i) to remove any false positives that might have been missed by the automated check and (ii) to classify the new SecGFP-GPI spots into two groups based on whether X-FP co-detection was observed or not (see S.I.). The data at the site of the spot were represented as the average fold change over the surrounding background, as a function of time (Fig. 1b, solid traces), and compared to the average fold change of arbitrary regions within the cell ('Random') (Fig. 1b, dashed traces). The profile obtained (pooled from multiple cells) represented a spatial and temporal profile of the X-FP at SecGFP-GPI endocytic sites.

Using the pH pulsing assay, we found that the rise in pH 7 SecGFP-GPI intensity occurred only ~3 s prior to vesicle

generation as opposed to nearly 40 s for CME, followed by monitoring SecGFP-TfR internalisation (ref. [4] and Supplementary Fig. 2c). Furthermore, we observed a poor correlation ($r = 0.3$) between pH 7 vs. pH 5 intensity per endosome, indicating that SecGFP-GPI is endocytosed without a major concentration in the endocytic vesicle (Fig. 1c). In comparison to CME, it

should be noted that we faced two main challenges, (i) lack of a cytoplasmic marker (for e.g., clathrin) for the endocytic site, (ii) lack of strong concentration of the cargo prior to the endocytic vesicle pinching. Regardless, the protocol developed provided a reliable real-time assay for studying the spatio-temporal dynamics of the internalisation of GPI-anchored proteins (see, S.I.).

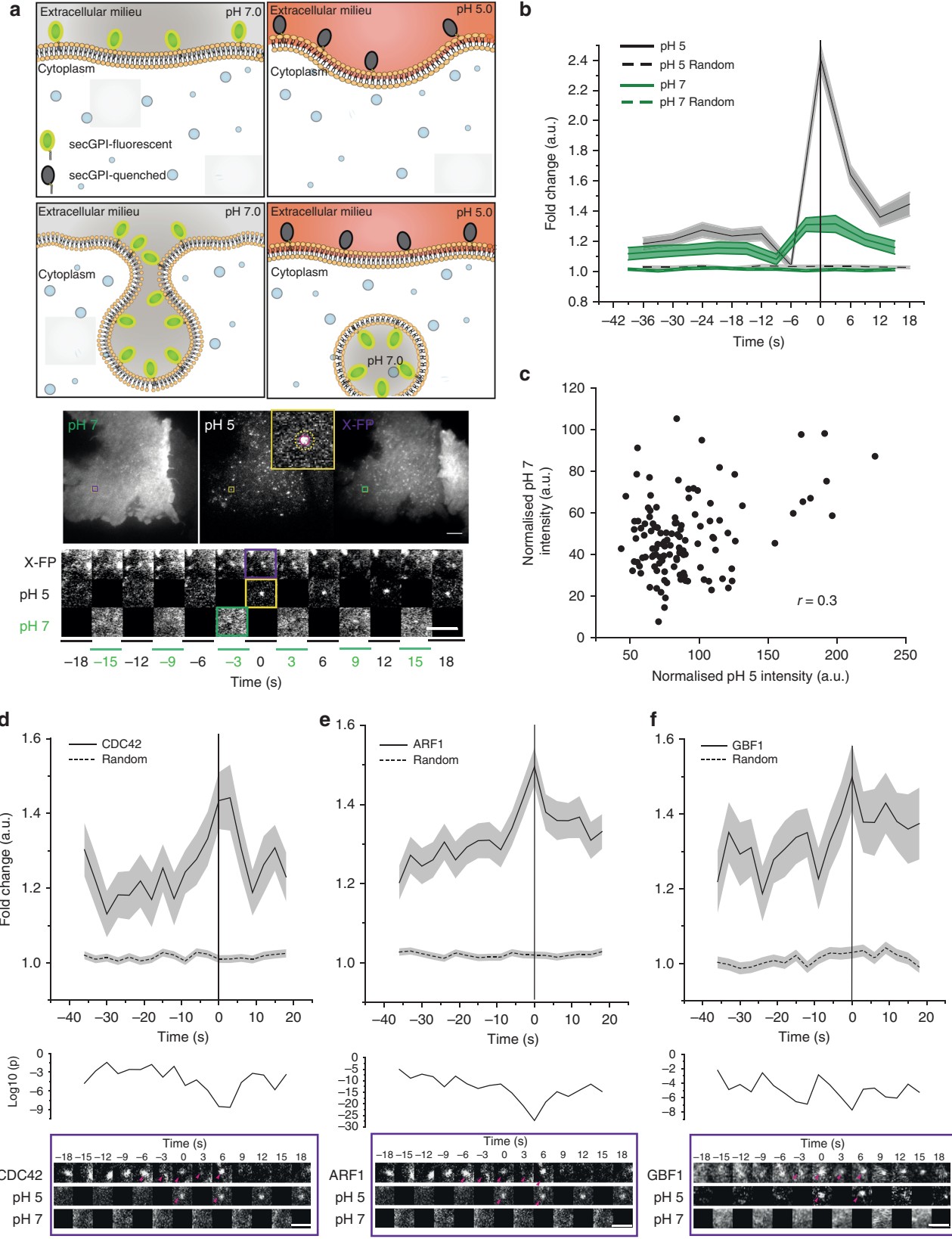

**GBF1, ARF1 and CDC42 are recruited to nascent CG endocytic sites.** We visualised the temporal dynamics of co-expressed CDC42 at the site of formation of nascent SecGFP-GPI endocytic vesicles. TagRFPt-CDC42 recruitment was quantified as average fold accumulation of CDC42 relative to the local background in all the endocytic events recorded over a 40 s time window straddling the endocytic event. A significant change in the intensity of CDC42 began at around −9 s and peaked around 0 to +3 s (Supplementary Fig. 2a). Based on the presence or absence of a co-detected CDC42 spot (within −18 to +18 s time window) at the endocytic site, we identified two populations via manual classification (see S.I. for a detailed description). We called them CDC42 Coloc (co-detection of CDC42 and SecGFP-GPI) and CDC42 NoColc (the remainder). We compared the fold accumulation of all the CDC42 spots (CDC42 All), CDC42 Coloc and CDC42 NoColoc with the Random. While the CDC42 Coloc profile was similar to that of CDC42 All, the CDC42 NoColoc profile was comparable to the Random (Supplementary Fig. 2b). Thus, the endocytic sites detected by our assay consisted of two populations wherein one fraction exhibited an accumulation of CDC42 while the second fraction failed to show a discernable accumulation. CDC42 Coloc corresponded to the 56% (of the CDC42 All) SecGFP-GPI endocytic sites. As the removal of the events which did not coincide with the presence of CDC42 did not alter CDC42 recruitment profile (compare, Fig. 1d and Supplementary Fig. 2a), they were discarded from further analysis. While the reasons for not detecting CDC42 at all endocytic events is a function of both the signal and noise in the data, it may also reflect a genuine lack of recruitment at some endocytic events (see S.I. for a detailed explanation). Henceforth, for all X-FPs, we report pH pulsing traces that were classified as co-detected with the SecGFP-GPI (see Methods, S.I. and Table 1, for methods and statistical details and analysis).

We next examined another CG pathway regulator, ARF1[14], and found that mCherry-ARF1 was already accumulated at the forming CG endocytic sites at −36 s and peaked at 0 s (Fig. 1e, Supplementary Fig. 6a and Table 1). Predictably, mCherry-GBF1, an ARF1-GEF[16], was also recruited to SecGFP-GPI spots (Fig. 1f, Supplementary Fig. 6b and Table 1). The temporal profile of GBF1 was correlated to ARF1 ($r = 0.6$, Table 2). When we extended the time window of observation for ARF1 and GBF1 further back in time (nearly 60 s before scission), we concluded that the accumulation of ARF1 and GBF1 at CG endocytic sites initiates as early as −54 s (Supplementary Fig. 2f–g). In absence of other molecules upstream of GBF1 and ARF1, this pair currently serves as the earliest initiators of the CG endocytic pathway. Furthermore, when we assessed the ultrastructure of newly formed fluid-filled endosomes by electron microscopy (EM)[23] in

cells treated with the small-molecule inhibitor of GBF1, LG186[33], the number of CLICs and fluid uptake was drastically reduced, whereas, CME-derived vesicles and uptake appeared relatively unaffected (Supplementary Fig. 1e–f). In contrast to ARF1, GBF1 and CDC42, we did not observe a frequent recruitment of clathrin and dynamin to the CG endocytic sites. The recruitment profile was comparable to random for >65% endocytic events for both mCherry-clathrin and dynamin (Supplementary Fig. 2d–e and Table 1). The remainder of the profiles exhibited high levels of clathrin and dynamin at SecGFP-GPI endocytic sites. Nevertheless, both fraction showed a temporal trend similar to that observed for random (Supplementary Fig. 2d–e, compare grey, blue and random traces). Conversely, mCherry-dynamin was recruited to SecGFP-TfR endocytic sites proximal to scission in 80% of the cases (Supplementary Fig. 2c) in agreement with a previous study[4].

Despite being responsible for the timed removal of CDC42 from the plasma membrane[14], ARF1 (and GBF1) was recruited long before the recruitment of CDC42. The pH pulsing analysis thus revealed a surprising facet of the recruitment of ARF1 (and GBF1), which suggested a CDC42-independent function(s) for ARF1 in CG endocytosis. Taken together, these results establish a reliable real-time imaging assay to follow newly formed CG endocytic vesicles containing SecGFP-GPI, correlated with recruitment of its known regulators, ARF1, GBF1 and CDC42.

**Identification of BAR domain proteins in CG endocytosis.** CLICs, visualised within 15 s of their formation, have a pleomorphic tubular appearance and lack a discernable protein coat when visualised by EM[21,23]. This prompted us to investigate

**Table 1 pH pulsing assay data set**

| Molecule | % Coloc | # Spots SecGPI | # Spots Random | # Cell and Experiment |
|---|---|---|---|---|
| CDC42 | 56 | 219 | 3428 | 17 and 6 |
| ARF1 | 62 | 411 | 4952 | 12 and 3 |
| GBF1 | 62 | 132 | 1917 | 6 and 2 |
| IRSp53 | 60 | 309 | 4439 | 7 and 3 |
| ARP3 | 48 | 170 | 3277 | 16 and 6 |
| Lifeact | 61 | 244 | 4277 | 7 and 3 |
| PICK1 | 77 | 121 | 1968 | 22 and 10 |
| N-WASP | 27 | 256 | 1827 | 9 and 3 |
| Clathrin | 30 | 437 | 1801 | 8 and 4 |
| Dynamin | 34 | 130 | 1866 | 10 and 4 |

See section titled pH pulsing analysis (Methods and S.I) for details

**Fig. 1** Identification of newly formed SecGFP-GPI endocytic vesicles using a pH pulsing assay. **a** Schematic (top panel) of the pH pulsing assay depicting the effect of pH on SecGFP-GPI fluorescence during endocytic vesicle budding. Note the fluorescence of SecGFP-GPI is retained at high pH (top and bottom left panel) or when exposed to low pH if sequestered in a newly formed endocytic vesicle (bottom right panel), and quenched only when exposed to low pH (top right panel). Sequential TIRF images of AGS cell co-expressing SecGFP-GPI and mCherry-ARF1 collected at pH 7, pH 5 and in the RFP channels (middle panel). Newly formed endocytic vesicles (inset) (identified as in Supplementary Figure 1c) are used to construct a single frame (yellow rectangle) of the montage depicted (bottom panel). **b** Average of the normalised fluorescence intensities of pH 5 and pH 7 traces at the site of newly formed SecGFP-GPI endocytic vesicles compared to their respective random traces constructed from 120 endocytic events (pH 5 and pH 7) and 3428 random spots, derived from 17 cells pooled from four independent experiments. **c** The graph shows the fold enrichment of fluorescence intensity over the local background of pH 5 vs. pH 7 at the time of formation of the endocytic vesicles (data from 1b). **d–f** Graphs show the average normalised fluorescence intensity vs. time traces for the recruitment of TagRFPt-CDC42 (**d**), mCherry-ARF1 (**e**) and mCherry-GBF1 (**f**) to the forming SecGFP-GPI endocytic sites, and its corresponding random intensity trace (n, Table 1). The random traces were derived from randomly assigned spots of the same radius as the endocytic regions, as detailed in S.I. Endocytic distribution at each time point was compared to the random distribution by Mann–Whitney U test, and the $\log_{10}$ (p) [$\log_{10}$ (0.05) is −1.3 and $\log_{10}$ (0.001) is −2.5] is plotted below each trace (**d–f**). Representative montages from the respective data sets are depicted below the graphs (**d–f**). Arrowheads indicate the newly formed endocytic vesicle. Error bars, s.e.m. (**b**, **d–f**). Scale bar, 1.5 μm (**a**, **d–f**)

## Table 2 Cross-correlation calculated for indicated traces

| Molecule pair | Time interval (s) | |
| --- | --- | --- |
| | (−36 s to −12 s) | Specific interval |
| GBF1 vs CDC42 | 0.5767 | |
| GBF1 vs LIFEACT | n.s. | |
| GBF1 vs ARP3 | n.s. | |
| GBF1 vs IRSp53 | n.s. | |
| ARF1 vs GBF1 | 0.6323 | |
| ARF1 vs CDC42 | 0.564 | |
| ARF1 vs IRSp53 | n.s. | |
| ARF1 vs ARP3 | 0.6347 | |
| ARF1 vs LIFEACT | 0.7478 | |
| CDC42 vs IRSp53 | 0.65[a,b] | 0.5698[a] (−33 s to 0 s) |
| CDC42 vs LIFEACT | 0.6087 | 0.54 (−36 s to +6 s) |
| CDC42 vs ARP3 | n.s. | |
| ARP3 vs IRSp53 | n.s. | 0.7739[a] (−9 s to +9 s) |
| ARP3 vs LIFEACT | n.s. | 0.6292 (−33 s to 0 s) |
| ARF1 vs PICK1 | n.s. | 0.5698 (−36 s to −12 s) |
| PICK1 vs IRSp53 | −0.5353 | |

Rest of the calculations are done with traces with spot radius = 3 pixels and background donut = 6–8 pixels
1 pixel = 84 nm
n.s. not significant $p$ value considered here is 0.05 calculated via $t$-statistic
[a]Performed for traces with spot radius = 2 pixels and background donut = 6–8 pixels
[b]Performed with one frame shift

the role of BAR domain proteins (BDPs) in CG endocytosis due to their capability to sense or stabilise membrane curvature. Additionally, BDPs contain domains that can bind lipids and/or regulate actin machinery, including RhoGTPases[30]. Thus, we performed a dsRNA screen in S2R+ cells[15,16] to identify the BDPs involved in CG endocytosis using GBF1 (*garz*) and GFP dsRNA as positive and negative control, respectively. The screen yielded 10 'hits' which affected fluid-phase uptake (Fig. 2a, b and Supplementary Fig. 3). Predictably, endophilin A required for dynamin-dependent CIE endocytosis of GPCRs and Shiga Toxin[9,10] was not selected, whereas, GRAF1[31] previously shown to be involved in CG endocytosis was identified as a 'hit'. The other hits, sorting nexin 6[34] and centaurinβ 1A[35] have been implicated in late stages of endosomal trafficking; CIP4[36] and NWK[37] being dynamin interactors were not pursued in this study. We focused, instead, on two classes of BDPs, MIM/CG32082 (I-BAR domain) and PICK1 (BAR domain), primarily due to their interactions with CDC42 and ARF1, respectively.

**IRSp53 is necessary for CG endocytosis**. IRSp53, the mammalian orthologue of CG32082[38], has been implicated in filopodia formation. IRSp53 has a multi-domain architecture consisting of I-BAR, CRIB, SH3 and PDZB domains. Using its SH3 domain, IRSp53 is known to interact with many actin regulatory proteins such as WASP-family verprolin-homologous protein 2 (WAVE2)[39–41], Mena/VASP (vasodilator-stimulated phosphoprotein)[42,43], Eps8[42,44–46], mDia[40] (Fig. 3a). Furthermore, the recruitment of

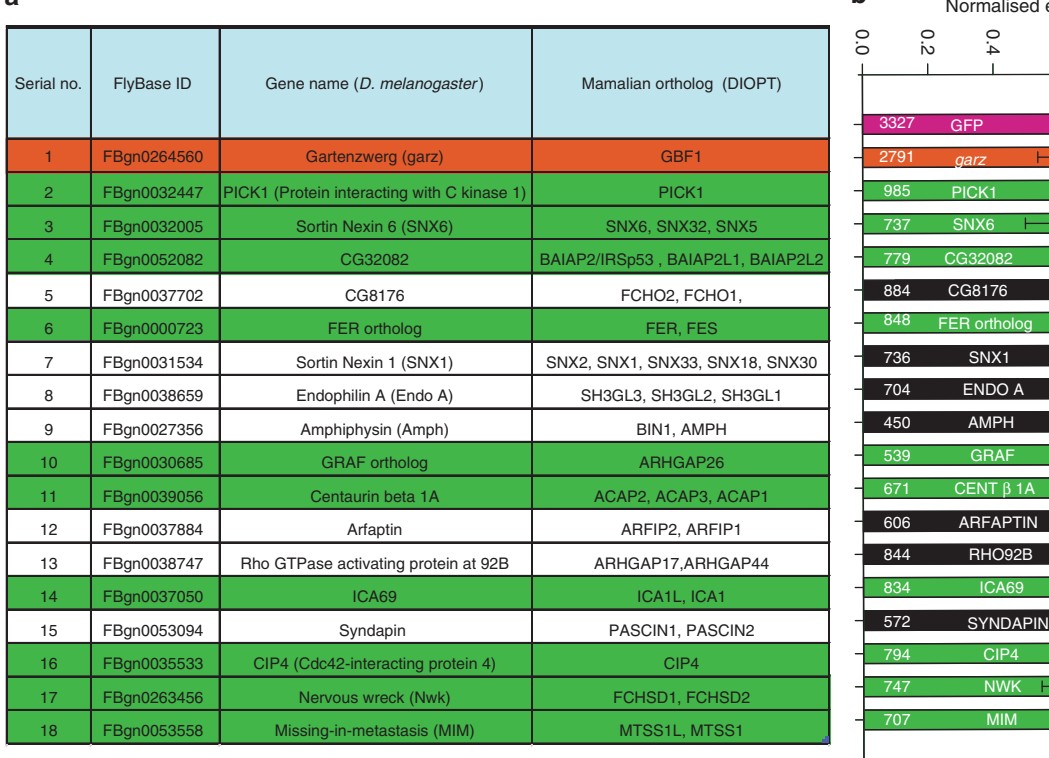

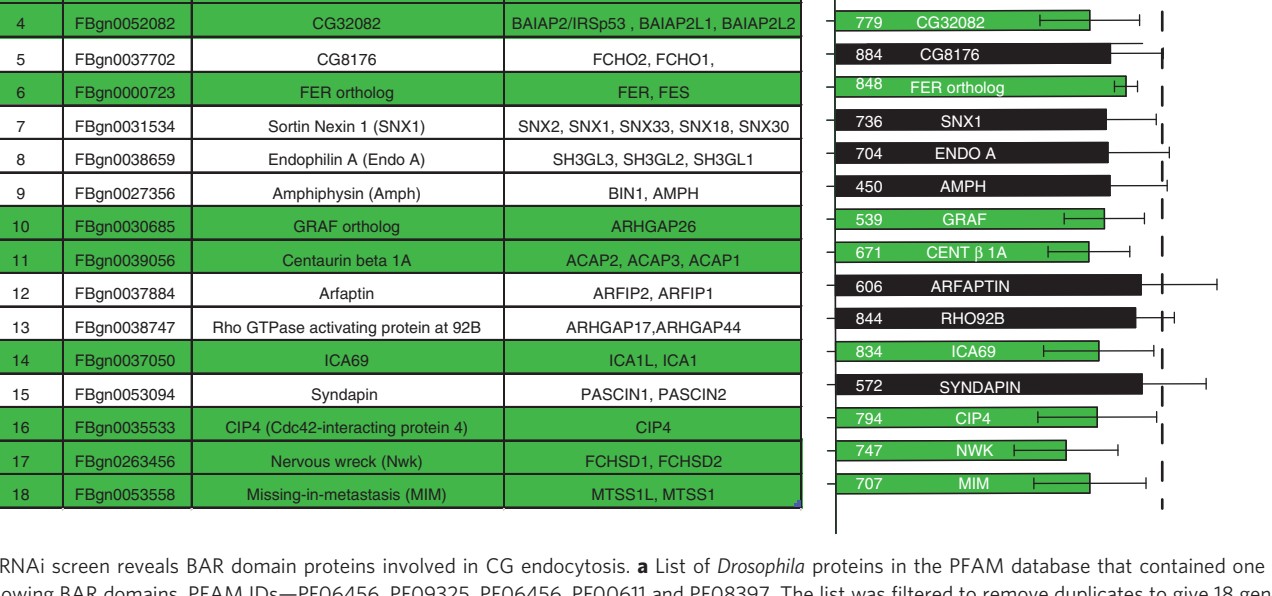

**Fig. 2** RNAi screen reveals BAR domain proteins involved in CG endocytosis. **a** List of *Drosophila* proteins in the PFAM database that contained one of the following BAR domains, PFAM IDs—PF06456, PF09325, PF06456, PF00611 and PF08397. The list was filtered to remove duplicates to give 18 genes. **b** The histogram shows normalised 5-min fluid-phase uptake in S2R+ cells treated with 10 μg of dsRNA for 4 days as indicated with dsRNA against GBF1 (*garz*) as positive, and GFP as negative controls. In a single experiment, mean uptake of one of GFP dsRNA coverslip was used to normalise the mean for rest of the coverslips. Data were pooled from three independent experiments and the cell numbers are indicated in the graph. The bars in green are significantly different from GFP dsRNA using two-sample $t$-test ($p$ value <0.05). Version 27 of the PFAM database was used to generate the list

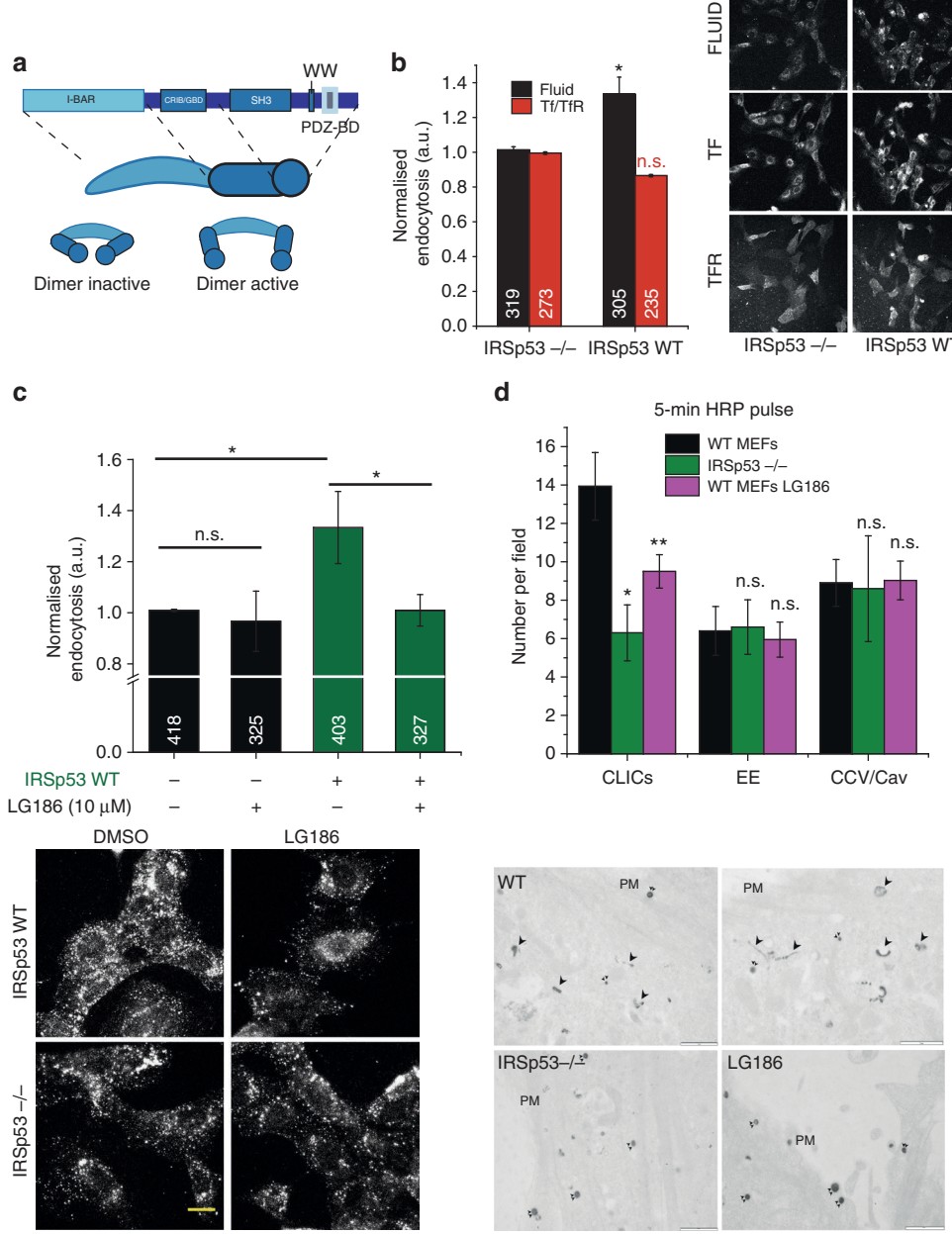

**Fig. 3** IRSp53 is involved in CG endocytosis. **a** Schematic depicting the domain organisation of IRSp53. IRSp53 exists in an inactive dimer state, which upon binding to GTP-CDC42 is activated, allowing SH3 domain to bind to its effectors. **b** Histogram (left) shows 5-min uptake of TfR and fluid phase in IRSp53 WT cells normalised to IRSp53−/− cells, along with representative images (right). Data were pooled from two independent experiments and the number of cells indicated in the figure. **c** Histogram (top) shows normalised 5-min fluid uptake in IRSp53−/− and IRSp53WT cells when treated with LG186 or vehicle (DMSO) along with representative images (bottom). Data were pooled from two independent experiments and the number of cells indicated in the figure. **d** Histogram (top) shows an average number of endocytic structures quantified per field from the electron microscope images (bottom). Data pooled from three independent blocks. Untreated WT MEFs (WT, top row), IRSp53 null MEFs (IRSp53−/−, bottom left) or LG186-treated WT MEFs (LG186, bottom right) were incubated for 5 min at 37 °C with 10 mg/ml HRP as a fluid-phase marker before processing for electron microscopy. Endocytic structures close to the plasma membrane (PM) are filled with the electron dense peroxidase precipitate. WT cells show a range of endocytic structures including vesicular structures (double arrowheads) and tubular/ring-shaped putative CLIC/GEECs (large arrowheads) but the IRSp53−/− cells and LG186-treated cells only show predominant labelling of vesicular profiles. $p$ value <0.05 (*), 0.001 (**) Mann–Whitney $U$ test (**b–c**) and two-sample Student's $t$ test (**d**). Error bars, s.d. (**b–d**). Scale bar, 20 μm (**b–c**), 1 μm (**d**), respectively. Schematic (**a**) was adapted with permission from MBInfo (www.mechanobio.info) Mechanobiology Institute, National University of Singapore

IRSp53 to the plasma membrane was compromised following ARF1 depletion[47]. Hence, IRSp53 was a good candidate to act as a signalling platform, linking CDC42 activation, membrane curvature and actin regulation for CG endocytosis.

To address the function of IRSp53 we compared endocytosis between mouse embryonic fibroblasts (MEFs) generated from IRSp53−/− mice (IRSp53−/− MEFs) and IRSp53−/− IRSp53WT addback MEFs (IRSp53WT MEFs)[42]. Loss of IRSp53 caused a significant reduction in fluid-phase uptake, without affecting TfR internalisation (Fig. 3b). We next addressed the nature of endocytosis in IRSp53−/− MEFs and found that the fluid-phase uptake in IRSp53−/− MEFs remained refractory to

LG186-mediated GBF1 inhibition (Fig. 3c). By contrast, GBF1 inhibition in IRSp53WT MEFs decreased fluid-phase uptake to the levels observed in IRSp53−/− MEFs (Fig. 3c) while endocytosed TfR remained unaffected (Supplementary Fig. 4a). We confirmed the lack of the CG endocytic pathway in IRSp53 −/− MEFs by ultrastructural analysis of endocytic structures marked by the fluid-phase marker, HRP using EM[14,23]. We observed a significant reduction of CLICs in IRSp53−/− MEFs when compared with WT MEFs, while the number of clathrin and caveolae-derived structures was relatively unaffected. Similar to Supplementary Fig. 1e, f, the CLICs were reduced significantly

upon LG186 treatment in WT MEFs as well (Fig. 3d and Supplementary Fig. 4b).

This led us to hypothesise that CG cargo would traffic via CME in the absence of IRSp53. Therefore, we looked at the trafficking of GPI-AP (GFP-GPI), fluid phase and TfR (CME) in the absence of IRSp53 at high resolution. We first counted the number of GFP-GPI and fluid endosomes, and found them to be significantly lower in IRSp53−/− MEFs relative to IRSp53WT MEFs (Fig. 4a). Conversely, TfR endosomal number was unaffected (Fig. 4a). We next, looked at co-localisation of GPI-AP/fluid phase with TfR. A relatively higher fraction of GFP-GPI

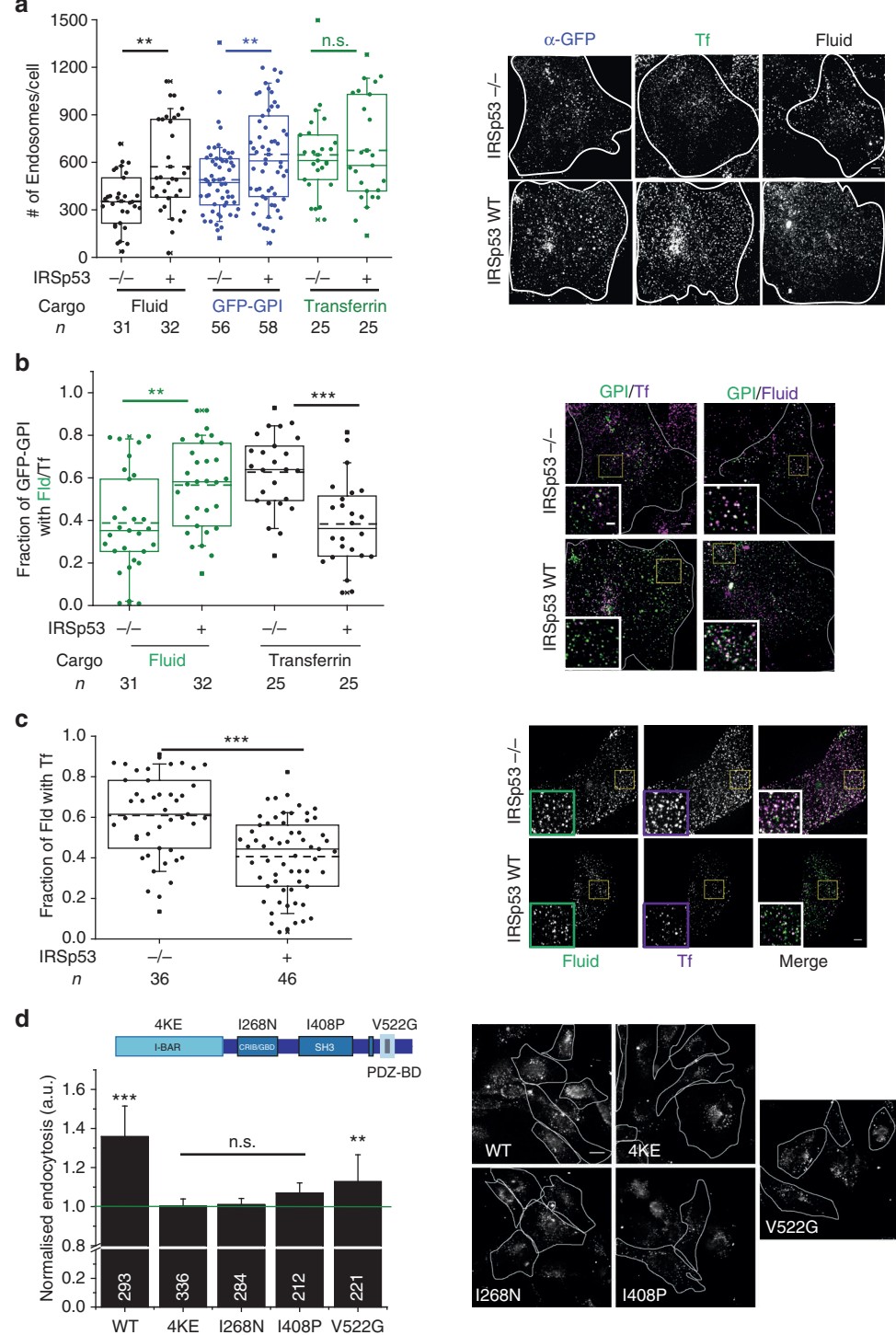

(Fig. 4b) and fluid phase (Fig. 4c) co-localised with co-pulsed Tf in IRSp53−/− MEFs than IRSp53WT MEFs. Thus, removal of IRSp53 specifically affected fluid phase and GPI-AP endocytosis, while CME remained unaffected. Moreover, in cells lacking IRSp53, GPI-AP and the residual fluid phase is endocytosed via CME.

We next analysed the contribution of different domains of IRSp53 on CG endocytosis by re-introducing into IRSp53−/− MEFs, GFP-tagged IRSp53WT and a number of mutants of IRSp53 specifically defective in various domains (Fig. 4d, schematic). We found that GFP-IRSp53WT and GFP-IRSp53V522G rescued endocytosis while the rest of the mutants failed to do so (Fig. 4d). In conclusion, these results indicated that IRSp53 is an essential and specific regulator of CG endocytosis that requires functional I-BAR, CRIB and SH3 domains.

**IRSp53 is recruited to forming CG endocytic vesicles**. The complete absence of CG endocytosis in IRSp53−/− led us to hypothesise that IRSp53 has a direct role to play in CG vesicle formation. Hence, we used pH pulsing assay and examined the recruitment of mCherry-IRSp53 to the forming SecGFP-GPI endocytic sites. A majority of (>60%) endocytic events exhibited prominent recruitment of IRSp53 (Fig. 5a, Supplementary Fig. 6c and Table 1). Since the I-BAR domain of IRSp53 has been shown to sense/induce negative curvature in a membrane tension and protein concentration-dependent manner[48], we looked at changes in its spatial distribution during the formation of endocytic vesicle using two types of masks—a spot and a ring mask (Fig. 5a, schematic). Unlike the intensity profiles of CDC42 that did not exhibit any differential temporal patterns of recruitment between the two types of masks (Fig. 5b, black vs. red trace), IRSp53 displayed a biphasic recruitment pattern (Fig. 5a). In phase I (−36 to −15 s), IRSp53 was first recruited over a large area indicated by an increase in intensity in both spot and ring traces. Subsequently, in phase II (−15 to 0 s), IRSp53 was confined at the centre indicated by a concerted decrease in the ring mask intensity and increase in the spot mask intensity trace (Fig. 5a, black vs. red trace). The increase of IRSp53 was more prominent in phase II toward the centre (Fig. 5a, black vs. purple trace) and correlated with the recruitment of CDC42 ($r = 0.6$; Table 2).

To visualise the recruitment of GFP-IRSp53 at high spatial resolution, we co-expressed a GBP-APEX reagent (GFP-binding protein soybean ascorbate peroxidase) and processed for EM as described previously[49] (Fig. 5c, Supplementary Fig. 4c and Supplementary Movies 3–4). GBP-APEX binds to GFP and converts 3,3′-diamino-benzamidine into an osmiophilic polymer in the presence of $H_2O_2$[49]. Images of 3D rendering from the electron densities revealed that IRSp53 associated with tubular structures characteristic of CLICs as described previously[23] and as

expected, with filopodial structures[38] (Fig. 5c, see Methods). In the 2D sections, IRSp53 was observed to accumulate as discrete patches at the plasma membrane (PM) (Supplementary Fig. 4c i–ii, arrowheads) and was frequently associated with tubulovesicular invaginations or tubular structures close to PM and filopodial tips (Supplementary Fig. 4c iii, double arrowheads).

We further validated recruitment of GFP-IRSp53 with an alternate technique, protein-retention expansion microscopy (ProExM)[50], a derivative of expansion microscopy[51]. Expansion microscopy enables imaging of diffraction-limited structures by physically separating them using a swellable polymer cross-linked with the cell. This allows multi-colour super-resolution imaging of a sample with conventional regents and microscope. Thus, structures of around 250 nm will be scaled to 1 µm in a 4× expanded sample (see Methods), providing a potential apparent resolution of around 70 nm[51] with conventional imaging technology. We stained IRSp53−/− GFP-IRSp53 MEFs for CD44 and IRSp53, processed the cells (Methods and ref. [50]), and imaged the samples using ×100 objective in spinning disk microscope [Supplementary Fig. 4e (1a–b) and Supplementary Movie 5]. In accordance with our EM images, we observed enrichment of IRSp53 at the tips of the filopodia [Supplementary Fig. 4e (2a–c) and Supplementary Movie 5]. Additionally, we could identify several invaginations of CD44 [Supplementary Fig. 4e (3–8, see arrowheads) and Supplementary Movie 5] and found enrichment of IRSp53. As the neck constriction is expected to develop [Supplementary Fig. 4e (3–4 and 6, arrowheads) and Supplementary Movie 5], we could see progressive enrichment of IRSp53 around the region of the constriction, relative to other regions of the invagination.

The pH pulsing trace showed a specific pattern of enrichment of IRSp53 over time. This is consistent with two scenarios: (1) that IRSp53 enrichment occurs specifically at the neck which constricts over time and, (2) that IRSp53 coats the entire tubule with higher enrichment around the necks which in turn constricts over time. Our interpretation of EM and proExM data support the latter scenario. Thus, the complete loss of CG endocytosis in IRSp53 null cells, and localisation of IRSp53 to forming CG endocytic vesicles suggests a role of IRSp53 in the vesicle scission.

**Branched actin nucleation is required for CG endocytosis**. A functional CG endocytic pathway requires dynamic actin since inhibition of actin polymerisation (Latrunculin A), or filament stabilisation (Jasplakinolide), inhibited CG endocytosis[13]. CDC42 and IRSp53 are core components of a signalling axis that indirectly controls the location and activity of the ARP2/3 actin nucleation complex[39,52,53]. More pertinent, CK666[54]-mediated inhibition of ARP2/3 complex, impaired both fluid-phase and TfR uptake in a dose-dependent manner. However, the extent of

**Fig. 4** CG pathway is abolished in the absence of IRSp53. **a** The box plot shows the number of endosomes per cells (left) for endocytosed GFP-GPI (α-GFP Fab), fluid phase and TfR in IRSp53−/− and IRSp53 WT cells when pulsed for 2 min along with representative images (right). Data were pooled from two independent experiments and the number of cells indicated below the graph. **b** Plot (left) shows quantification of the fraction of GFP-GPI endocytic vesicles containing fluid phase or Tf. Images (right) show representative single confocal slices of a 2-min pulse of α-GFP Fab (green)/TMR-Dextran (magenta) and α-GFP Fab (green)/A568-Tf (magenta) in IRSp53−/− (top row) and IRSp53WT (bottom row) cells. The inset depicts magnified views of the indicated region; single-channel images are in panel 4a. Data were pooled from two independent experiments and the number of cells is indicated below the graph. **c** Plot (left) showing quantification of the fraction of 1-min fluid-phase endocytic vesicles containing Tf. Images show representative single confocal slices of a 1-min pulse of TMR-Dextran (green) and A647-Tf (magenta) in IRSp53−/− (top row) and IRSp53WT (bottom row) cells. Inset depicts magnified views of the indicated region. Data were pooled from two independent experiments and the number of cells indicated below the graph. **d** Histogram (left) shows 5-min uptake of fluid phase in IRSp53−/− MEFs transduced with virus expressing GFP-IRSp53 WT, GFP-IRSp53 4KE, GFP-IRSp53 I268N, GFP-IRSp53 I408P and GFP-IRSp53 V522G, normalised to that in IRSp53−/− MEFs, along with representative images (right). Data were pooled from two independent experiments and the number of cells indicated in figure except for IRSp53−/− (381). *p* value <0.01 (*) and 0.001(**) by Mann–Whitney *U* test (**a–d**). Scale bar, 20 µm (**d**), 5 µm (**a–c**), respectively. Error bars (**d**) represent s.d.

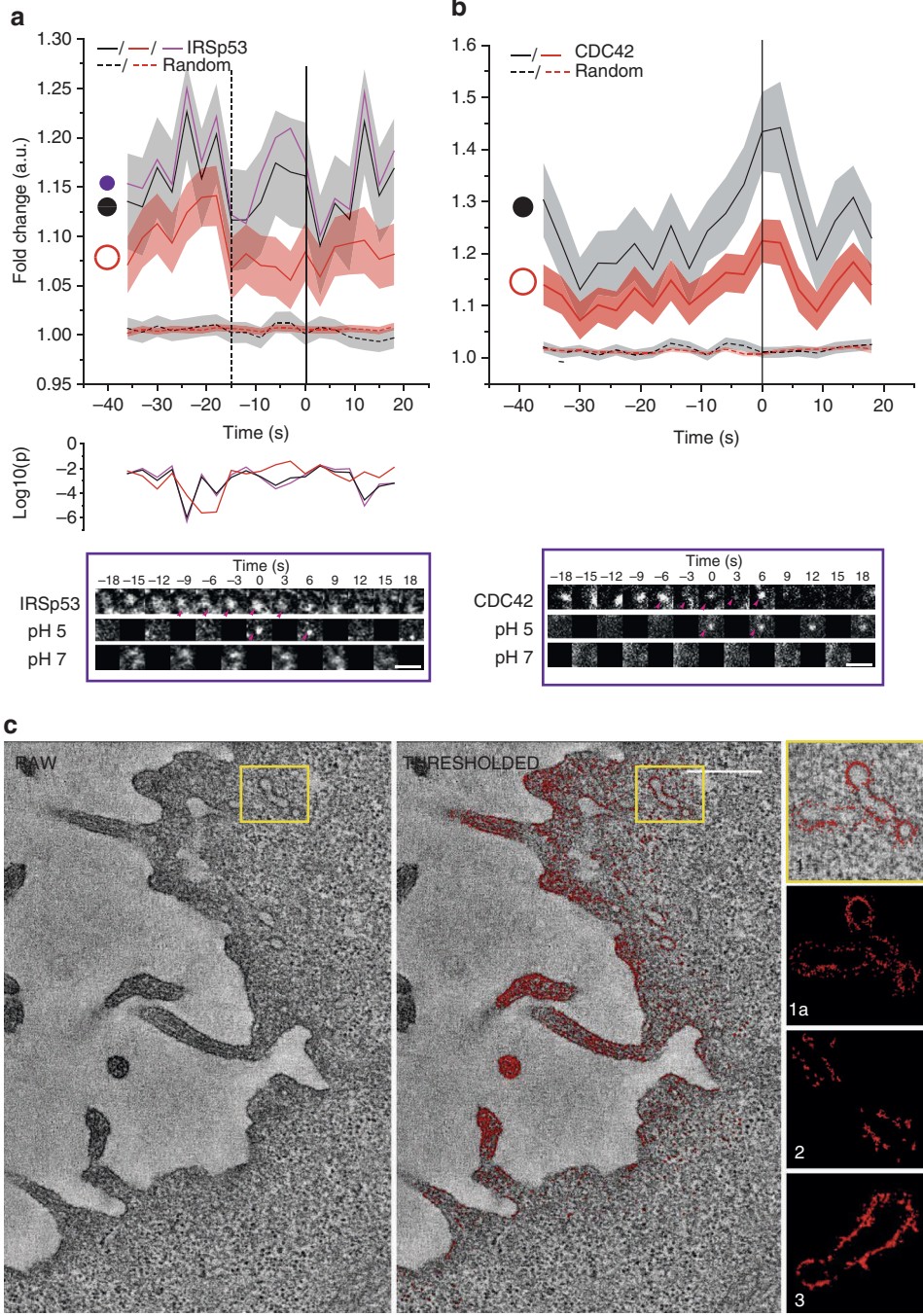

**Fig. 5** IRSp53 is recruited to forming CG endosomes. **a** Graphs show the average normalised fluorescence intensity vs. time traces for the recruitment of three different regions [circles, violet, $r = 170$ nm and black ($r = 250$ nm) and annulus, orange ($r = 250–420$ nm)] for the recruitment of IRSp53-mCherry to the forming SecGFP-GPI endocytic sites and its corresponding random intensity trace ($n$, Table 1). **b** Graphs show the average normalised fluorescence intensity vs. time traces for the recruitment of TagRFPt-CDC42 to the forming SecGFP-GPl endocytic sites and its corresponding random intensity trace to two different regions [circle, black, $r = 250$ nm; and annulus, orange ($r = 250–420$ nm)]. Error bars, (**a–b**) represent s.e.m. ($n$, Table 1). The random traces were derived from randomly assigned spots of the same radius as the endocytic regions, as detailed in S.I. Endocytic distribution at each time point was compared to the random distribution (**a**) by Mann–Whitney $U$ test and the $\log_{10}$ ($p$) is plotted below each trace [$\log_{10}$ (0.05) is $-1.3$ and $\log_{10}$ (0.001) is $-2.5$]. Representative montages are depicted below the graphs (**a–b**). Arrowheads indicate the newly formed endocytic vesicle. **c** Electron micrographs of AGS cells co-transfected-GFP-IRSp53 and GFP-binding protein coupled to Apex (GBP-Apex). The DAB reaction was performed and the cells were processed for electron tomography. A single section of the original tomogram (left) and density-based thresholded of the same plane (middle) reveal electron dense structures containing IRSp53 at membrane surfaces. The whole of PM of the tomographic volume was rendered and different examples of enlarged tubular regions of interest show GFP-IRSp53 recruitment patterns (right). Scale bar, 1.5 µm (**a–b**) and 0.5 µm (**c**), respectively

inhibition of CG internalisation was markedly more prominent (Fig. 6a). By contrast SMIFH2[55]-mediated inhibition of formins, failed to inhibit CG endocytosis (Supplementary Fig. 5a).

We next explored the spatio-temporal dynamics of F-actin and ARP2/3 complex using pRuby-lifeact[56] and mCherry-ARP3, respectively, during the formation of the endocytic vesicle. ARP3 recruitment (Fig. 6b, Supplementary Fig. 6d and Table 1) began earlier than −35 s and peaked at −6 s. This was unexpected

since CDC42, a key regulator of ARP2/3[57] was not recruited until −9 s. Instead, the ARP3 profile was correlated with IRSp53 between −9 s and +9 s ($r = 0.8$; Table 2). This indicated that, at least initially, the ARP2/3 complex might be recruited in a CDC42-independent manner. On the other hand, F-actin accumulation began around −9 s and continued even after the scission event (Fig. 6c, Supplementary Fig. 6e and Table 1), highly correlated to the CDC42 profile ($r = 0.6$; Table 2). Thus, F-actin

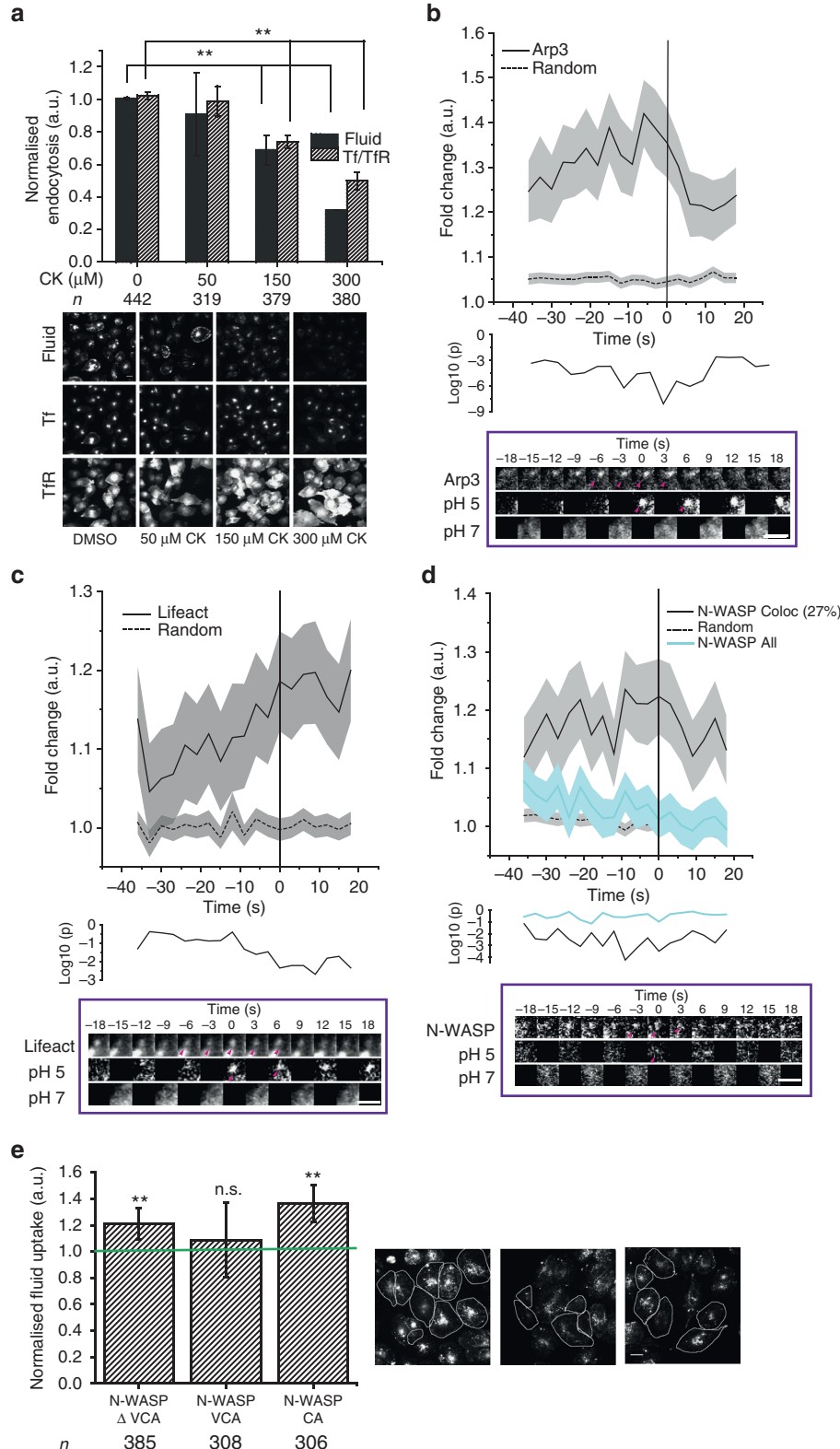

was generated coincident with the recruitment of CDC42, a known regulator of actin polymerisation[57]. These observations suggested that ARP2/3 complex might be first recruited in an inactive state, and then activated following the arrival of CDC42.

**ARP2/3 is inhibited by PICK1 at nascent CG endocytic sites**. To address how ARP2/3 complex was perhaps maintained at the forming endocytic pit in an inactive state, we analysed the role of PICK1, another 'hit' in the screen (Fig. 2). PICK1, a highly conserved protein, possesses PDZ and BAR domain (Fig. 7a) that by intra-molecular interaction, maintains PICK1 in an auto-inhibited state. This auto-inhibited state is further stabilised upon GTP–ARF1 binding to the PDZ domain[58]. Additionally, activated PICK1 negatively regulates ARP2/3-mediated actin polymerisation[58–60]. The ability of PICK1 to inhibit ARP2/3 complex is suppressed by GTP–ARF1[58]. To confirm a role of PICK1 in CG endocytosis in mammalian cells, we utilised a specific small-molecule inhibitor of PICK1, FSC231[61]. CG endocytosis (fluid phase) was inhibited in a dose-dependent fashion (Fig. 7b) by this inhibitor. Additionally, in stable PICK1 knockdown lines, fluid phase and folate receptor (FR-GPI, another GPI-AP) endosomal number were lower than that measured in scrambled shRNA stable lines, while TfR endocytosis remained unaffected (Fig. 7c, d).

Predictably, GFP-ARF1 and TagRFP-PICK1 co-localised in punctate spots at the TIRF plane in accordance with previous reports[58] (Supplementary Fig. 5b). To test the effect of ARF1 activity on PICK1 recruitment we co-expressed TagRFP-PICK1 with either ARF1-WT or dominant-negative (ARF1-DN; T31N) and active mutant (ARF1-DA; Q71L) and tracked the number and dynamics of PICK1 spots using TIRF microscopy (see Methods). The residence time of PICK1 increased significantly in the presence of ARF1-DN while it was reduced in the presence of ARF1-DA (Fig. 7e). ARF1-DN and DA have been shown to decrease and increase endocytosis, respectively[14]. Thus, the local presence of ARF1-GTP resulted in the removal of PICK1 from the membrane. Thus, we hypothesised that PICK1 was recruited to forming CG endosome at the early stage, rendering ARP2/3 inactive.

To explore this possibility, we utilised the pH pulsing assay to visualise PICK1 at the CG endocytic site. We found that TagRFP-PICK1 was recruited to the forming CG endocytic sites (Fig. 7f, Supplementary Fig. 6f and Table 1) in a pulsatile fashion. Maximum enrichment occurred at −12 s, with an eventual loss corresponding to the time of the rapid rise in ARF1 recruitment around −9 s (Fig. 1e, $r = 0.6$, Table 2).

Thus, the pH pulsing assay has led to the realisation that in CG endocytosis, the interplay of two BDPs, PICK1 and IRSp53, regulate ARP2/3 complex. These BDPs interact with the ARP2/3 complex (Supplementary Fig. 5b, upper panel and Supplementary Fig. 4d) regulate its activity at the forming CG endocytic sites, in opposing fashion under the influence of ARF1 and CDC42.

## Discussion

CG endocytosis was initially discovered as a route for the entry of toxins[62], fluid phase and GPI-anchored proteins when CME was perturbed[11,12,63], raising some concerns regarding its physiological role in unperturbed cells[18]. Using a pH pulsing assay we show, here, that a majority of SecGFP-GPI-containing endocytic vesicles form due to a stereotypical and temporally orchestrated recruitment of the key molecular machinery namely, CDC42, ARF1 and GBF1, which, in turn, mediate the coordinated assembly of specific BAR-containing, membrane deforming and actin regulatory proteins, IRSp53 and PICK1. Notably, however, the vast majority of the endocytic vesicles are devoid of clathrin and dynamin. Thus, a dedicated complex protein machinery drives CG internalisation, similarly to that being observed in CME. The ability of the pH pulsing assay to provide a temporal profile for the recruitment dynamics of the molecular players has considerably extended our understanding of CG endocytic vesicle formation.

The GBF1/ARF1 pair is the earliest module to be assembled and judging by their recruitment profiles, it takes around 1 min to assemble the molecular machinery for CG endocytosis. How this pair is concentrated at a forming CG endocytic site is an open question. The CG machinery includes CDC42, ARP2/3 and F-actin along with BDPs, PICK1 and IRSp53, and in the model (Fig. 8) we propose a biphasic mechanism correlated with the ARF1 recruitment kinetics. In the first phase, the accumulation is slow, accompanied by the presence of PICK1 and absence of CDC42. In the second phase, beginning around −9 s (before scission), the accumulation of ARF1 speeds up concomitant with the arrival of CDC42 and loss of PICK1. How the kinetics of ARF1 recruitment is regulated is unclear since its GAP (GTPase-activating protein) is presently unknown, as is the GEF for CDC42. Nevertheless, the presence of PICK1 provides an explanation behind ARP2/3 recruitment in an inactive state to the forming CG endocytic vesicles long before CDC42. ARP2/3 is then induced to promote actin branching only upon the arrival of IRSp53 and CDC42.

The role of ARP2/3 in CG endocytosis is reminiscent of the endocytic process occurring in the budding yeast. In this system, the endocytic machinery strictly depends on Las17, the yeast homologue of N-WASP but not so much on clathrin and dynamin[64,65]. There are however important differences. In CG endocytosis, the ARP2/3 complex appears to be activated independent of its canonical NPF, N-WASP, a CDC42 effector[57]. First, not only did N-WASP fail to recruit to forming CG endocytic sites (Fig. 6d and Supplementary Fig. 6g), over-expression its dominant-negative mutants also failed to inhibit CG endocytosis (Fig. 6e). By contrast, in CME, both ARP2/3 complex and N-WASP are recruited to budding CME vesicles, and influence endocytosis in some cell types[28].

The unexpected recruitment profile of ARP2/3 and the identification of two BDPs, PICK1 and IRSp53 as upstream regulators

**Fig. 6** Arp2/3-based actin machinery is required for CG endocytosis. **a** Histograms (top) show quantification of fluid phase and TfR uptake in AGS cells treated with DMSO alone (0 μM) or the indicated concentrations of ARP2/3 inhibitor, CK666, normalised to DMSO-treated controls, along with its representative images (below). Data are pooled from two independent experiments and the number of cells shown indicated the graph. **b–d** Graphs show the average normalised fluorescence intensity vs. time traces for the recruitment of mCherry-ARP3 (**b**), pRuby-Lifeact (**c**) and mCherry-NWASP (**d**) to the forming SecGFP-GPI endocytic sites, and its corresponding random intensity trace (n, Table 1). The random traces were derived from randomly assigned spots of the same radius as the endocytic regions, as detailed in S.I. Endocytic distribution at each time point was compared to the random distribution by Mann–Whitney U test and the $\log_{10}$ (p) [$\log_{10}$ (0.05) is −1.3 and $\log_{10}$ (0.001) is −2.5] is plotted below each trace (**b–d**). Representative montages are depicted below the graphs. Arrowheads indicate the newly formed endocytic vesicle. **e** Histogram (left) shows normalised 5-min mean fluid-phase uptake in AGS cells overexpressing pIRES-CA domain, GFP-VCA domain and GFP-N-WASPΔVCA from N-WASP compared to un-transfected cells and representative images (right). The transfected cells are outlined. Data were pooled from two independent experiments and the number of cells shown below the graph. Error bars represent s.e.m. (**b–d**) and s.d. (**a, e**). p value <0.01 (\*), and 0.001 (\*\*) by Mann–Whitney U test (**a, e**). Scale bar, 1.5 μm (**b–d**), 20 μm (**a, e**)

of ARP2/3 activity also suggests a reason for this biphasic mechanism. PICK1 operates in the early phases, and may function as an inhibitor of ARP2/3, consistent with the modest recruitment of F-actin in the presence of PICK1 observed here and as reported previously[59,60]. PICK1 recruitment occurs via its BAR domain since when mutated and overexpressed, it acts as a dominant negative for CG endocytosis (Supplementary Fig. 5c). PICK1 recruitment is not only rapid, but also transient. We find that the localisation of PICK1 to the membrane was negatively correlated to the activity of ARF1, similar to that observed in neuronal cells[58], wherein GTP–ARF1 interaction with PICK1 rendered PICK1 incapable of inhibiting ARP2/3 complex. This is

followed by the second phase characterised by the simultaneous recruitment of CDC42/IRSp53 effector complex enabling the activation of ARP2/3 and subsequent polymerisation of actin at the site of endocytosis. However, the NPF linking IRSp53 and ARP2/3 activation is not yet characterised and is the subject of investigation.

There is no single unifying theme for vesicle scission in CIE and multiple modules may co-exist. Recently, endophilin A has been shown to facilitate tubule scission by a combination of scaffolding, dynamin recruitment and dynein-mediated elongation of membrane tubules leading to an increase in friction[66]. In the CG pathway, IRSp53 emerges as a major player. This protein

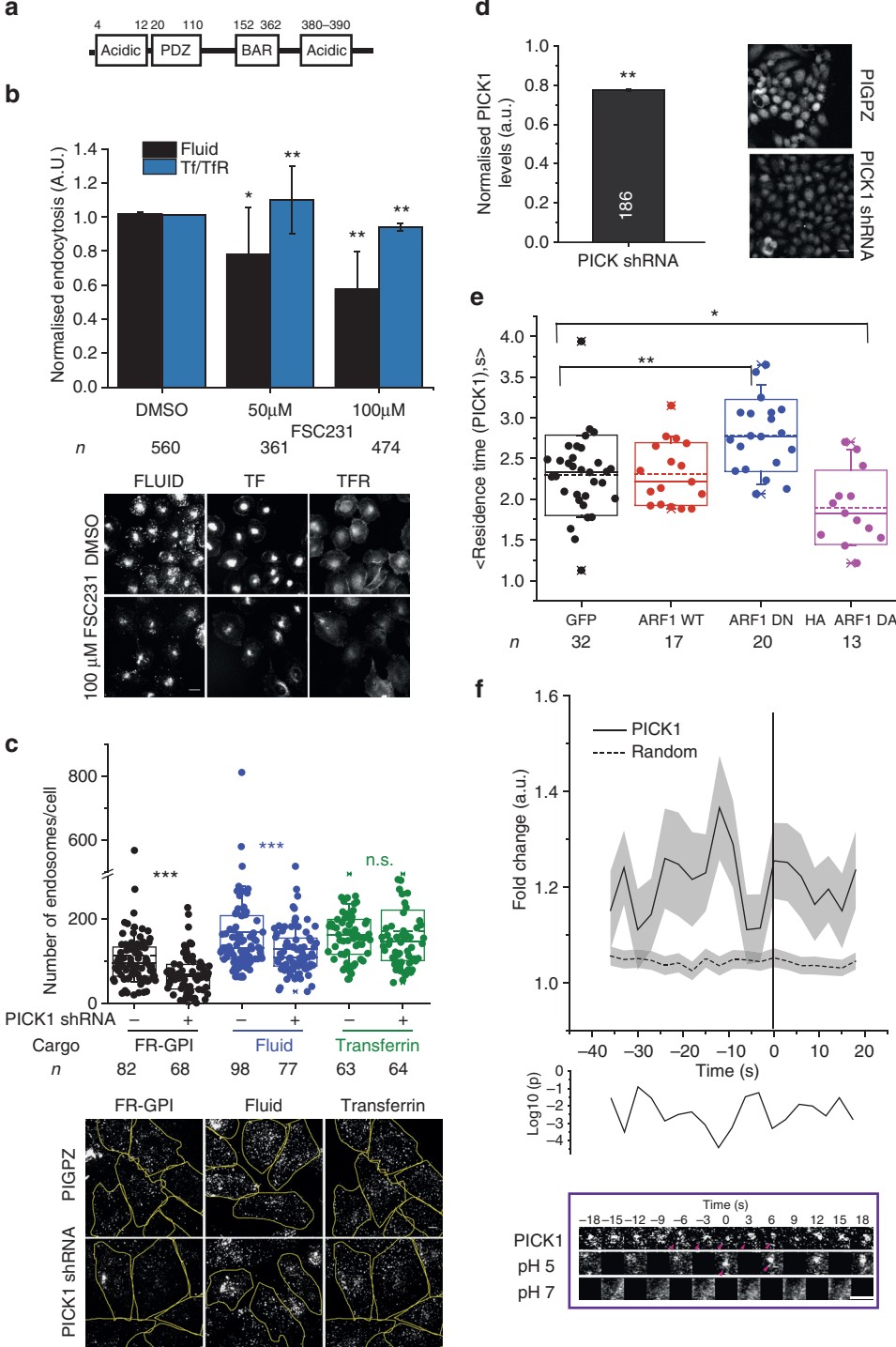

may function by multiple mechanisms: it can couple negative curvature with membrane tension, it can scaffold membrane at moderate densities[48], and it can regulate actin polymerisation as it does in filopodia formation[38]. A minimal model accounting for all these activities suggests that IRSp53 might be enriched at the vesicle neck, where it would regulate the actin machinery necessary to trigger CG vesicle scission. The spatio-temporal dynamics and ultrastructure analysis of IRSp53 recruitment at CG sites are consistent with such a model. However, the data does not permit an unequivocal picture; better resolution and reagents are necessary to verify this speculation. Lastly, the complete and specific loss of CG endocytosis (but not CME) in the absence of IRSp53 makes the requirement for IRSp53 necessary for CG endocytic process.

In summary, we propose that CG endocytic vesicle formation begins with GBF1/ARF1 concentrating at sites of endocytic pits. Following this, though ARP2/3 is recruited, it is held in an inactive state by PICK1 (Fig. 8a). Meanwhile, IRSp53 is recruited (potentially via its I-BAR domain) and activated by CDC42 leading to ARP2/3 activation via unknown effector(s) (Fig. 8b). The loss of IRSp53 and ARP2/3 from the membrane as the endocytic vesicle is pinched is consistent with their role in endosomal neck dynamics, providing a new candidate for molecular machinery of the pinching process in the absence of dynamin. It is conceivable that the IRSp53 provides a scaffold for a friction-based scission mechanism as recently suggested[66], with actin polymerisation providing driving force for tubule elongation. Alternatively, this force could arise from the involvement of a microtubule-based machinery as recently advocated in the internalisation of cholera toxin[67]. The assays developed here and the identification of a number of molecular players and their temporal recruitment profile provides a path towards understanding a molecular mechanism for the formation of a CG endocytic vesicle.

## Methods

**Cell culture, reagents, and plasmids**. See Supplementary Information.

**pH pulsing assay**. pH pulsing assay was adapted for Nikon TE 2000 TIRF microscope (for the details of the microscope, see microscope section in S.I.) from a similar assay as described previously[4]. Briefly, FR-AGS cells were plated on custom-designed coverslip bottom dishes and were transfected with SecGFP-GPI and X-FP constructs 12–14 h before the assay. The dishes were then fitted with a custom-designed holder to place inlet and outlet tubing. Tubing from a pH 7.4 (HEPES) and pH 5.5 (MES) buffers kept in a water bath at 38 °C went through a peristaltic flow controller (Bioscience Tools) into the cell chamber. The temperature of the cells in the dish was maintained at 30 °C by maintaining the buffers, the objective and the microscope (using a chamber) at appropriately high temperatures. Imaging at 30 °C slows the endocytic process to match the time resolution

achieved by the flow setup (3 s). Images were captured using a script written in open source imaging software, Micromanager, to control the time of flow and imaging. Typically, buffers are exchanged every 3 s and three images are collected sequentially before the end of 3 s in two channels, GFP and RFP, using camera exposure of 100 ms. The chamber around the microscope was built with the help of NCBS Mechanical Workshop, AC department and Dr Manoj Matthew (NCBS-CIFF).

**Endocytic assay**. Mammalian endocytic assays: All population-based endocytic assays were performed as described[14]. For endocytic assays in mammalian cells, 2-day plated cells on poly-D-Lysine-coated dishes were used. The assays were performed in the water bath maintained at 37 °C. The media was removed and replaced with media containing fluorescently labelled probes at appropriate dilutions for the required time (TMR-Dextran was used at 1 mg/ml, α-GFP was used at 20 µg/ml, Cy3-Mov18 was used at 5 µg/ml and Tf was used at 10 µg/ml). The cells were then transferred to ice and washed with ice-cold medium 1 buffer (140 mM NaCl, 20 mM HEPES, 1 mM $CaCl_2$, 1 mM $MgCl_2$, 5 mM KCl, pH 7.4). The cells were then stripped for surface-bound Tf with ascorbate buffer (160 mM sodium ascorbate, 40 mM ascorbic acid, 1 mM $MgCl_2$, 1 mM $CaCl_2$, pH 4.5). In the case of GPI-AP (GFP-GPI and FR-GPI), the surface was removed using PI-PLC which cleaves GPI anchor[14]. Cells were treated with PI-PLC (50 µg/ml) for 1 h on ice. The cells were fixed with 2.5% paraformaldehyde and stained for surface TfR.

BAR domain screen: RNAi screen for BDPs in *Drosophila* genome was done on $S2R^+$ cells stably expressing TfR[15]. Briefly, the cells were plated in a 12-well plate (0.5 million cells/well) for 1 h. The media was then replaced with 600 µl of serum-free media supplemented with appropriate dsRNA at (final amount, 10 µg) for 1 h, post which 600 µl of serum containing media was added. After 4 days of depletion, the cells were assayed for endocytosis. On the 4th day, cells were deadhered from the well by manual pipetting and plated on coverslip bottom dishes. Cells were pulsed with TMR-dextran diluted in serum containing media for 5 min. The cells were subsequently transferred to ice and washed with ice-cold medium 1 buffer (supplemented with 1 mg/ml BSA and glucose). The cells were then fixed using 2% paraformaldehyde (5 min on ice and 15 min at room temperature). dsRNA was prepared from the *Drosophila* Open Biosystems library v1[15].

HRP uptake and electron microscopy: WT and IRSp53−/− MEFs were serum starved for 45 min in the presence or absence of 10 µM LG186 compound. Cells were treated for 2 or 5 min at 37 °C with 10 mg/ml HRP in serum-free medium, rapidly washed with complete medium on ice and subsequently fixed with 2.5% glutaraldehyde in PBS. Peroxidase development and further processing were performed as described previously[23]. Cells were sectioned parallel to the substratum and viewed unstained on a Jeol 1011 (Tokyo, Japan) transmission electron microscope equipped with a Morada Soft Imaging camera (Olympus) at two-fold binning. Quantitation was performed as follows: for 5-min HRP uptake, images were captured at random across the monolayer by moving a defined distance across the grid to avoid user bias. For 2-min uptake, whole cells positive for HRP were imaged and montaged to generate a high-resolution image encompassing the entire cell. Five cells were imaged and montaged for each replicate. HRP-labelled elements per image or per cell profile were classified using the following criteria; vesicular elements including clathrin-coated vesicles and caveolae—circular profiles <200 nm in diameter; early endosomes–circular and ring-shaped profiles, including multivesicular structures >200 nm in diameter; CLIC/GEEC–other profiles including tubules and small ring-shaped structures <200 nm in diameter.

**Ultrastructural localisation of IRSp53**. Thin sections: Localisation of GFP-tagged IRSp53 was performed as described previously[49]. Cells (FR-AGS) were seeded onto 30 mm tissue culture dishes (TRP) and transfected 24 h later using Lipofectamine

**Fig. 7** PICK1 is involved in CG endocytosis and is negatively regulated by ARF1. **a** Schematic depicts domain organisation of PICK1. **b** Histograms (top) show quantification of fluid-phase and TfR uptake in AGS cells treated with DMSO alone (0 µM) or the indicated concentrations of PICK1 inhibitor, FSC231, normalised to DMSO-treated controls, along with its representative images (below). Data were pooled from two independent experiments with the cell numbers shown below the graph. **c** Box plot (top) shows the number of endosomes per cells for FR-GPI (Cy3-Mov18), fluid phase and TfR in scrambled (PIGPZ) and PICK1 shRNA-infected AGS cells when pulsed for 2 min along with representative images (bottom). Data are pooled from two independent experiments and the number of cells indicated below the graph. **d** Histogram (left) shows normalised PICK1 levels measured by immunostaining in PICK1 shRNA-infected AGS cells along with representative images (right). Data were pooled from two independent experiments with the cell numbers indicated in the figure except for PIGPZ (292). **e** Box plot (top) shows the residence time of TagRFPt-PICK1 spots at the TIRF plane (see Methods), averaged in an individual cell expressing either GFP, GFP-ARF1 WT, GFP-ARF1 DN or HA-ARF1 DA. The data are pooled from two independent experiments with cell number indicated below the graph. **f** The graph shows the average normalised fluorescence intensity vs. time trace for the recruitment of TagRFPt-PICK1 to the forming SecGFP-GPI endocytic sites and its corresponding random intensity trace (*n*, Table 1). The random traces were derived from randomly assigned spots of the same radius as the endocytic regions, as detailed in S.I. Endocytic distribution at each time point was compared to the random distribution by Mann–Whitney *U* test and the $\log_{10}$ (*p*) [$\log_{10}$ (0.05) is −1.3 and $\log_{10}$ (0.001) is −2.5] is plotted below. Representative montage is depicted below. Arrowheads indicate the newly formed endocytic vesicle. Error bars represent s.e.m. (**f**) and s.d. (**b**, **d**). *p* value <0.01 (*), 0.001(**) and 0.0001 (***) by Mann–Whitney *U* test (**b**–**e**). Scale bar, 1.5 µm (**f**), 20 µm (**b**, **d**) and 5 µm (**c**)

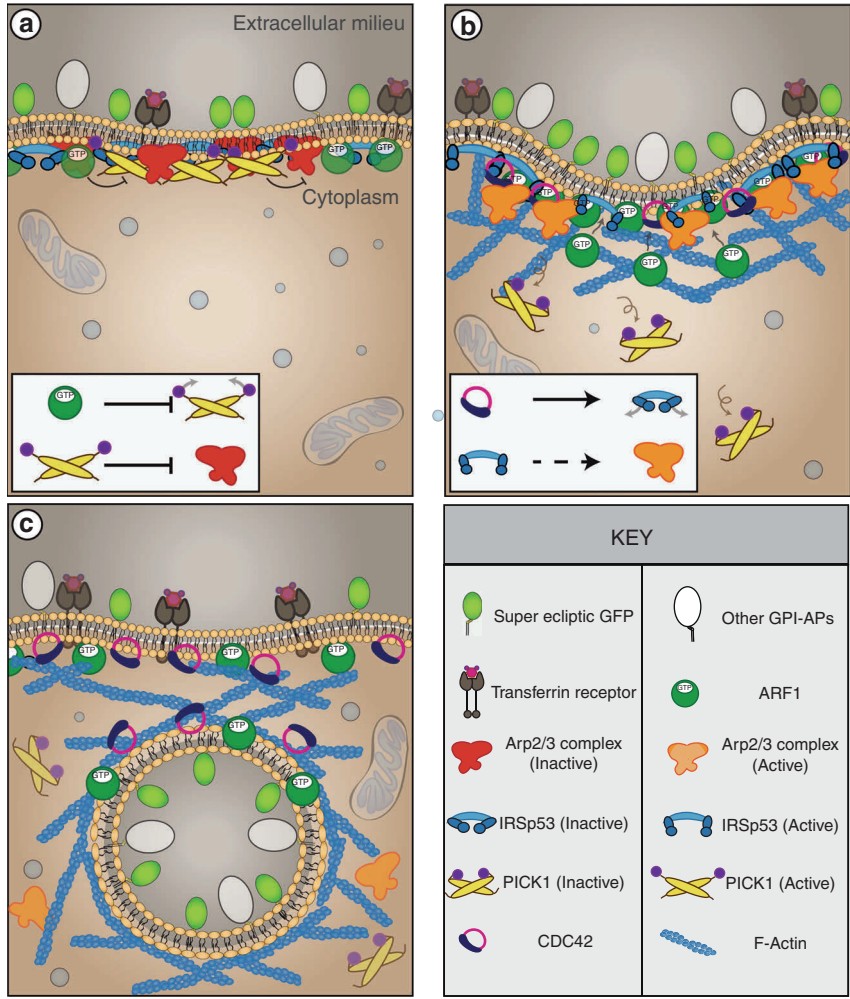

**Fig. 8** Schematic depicting the proposed biphasic mechanism for CG endocytic vesicle formation. **a** Phase I: Characterised by the recruitment of ARF1/GBF1, PICK1, ARP2/3 and IRSp53 but not the buildup of F-actin and CDC42. Here, IRSp53 may be recruited by its I-BAR domain in the absence of GTP-CDC42, keeping its SH3 domain in an intra-molecular inhibited state. PICK1 keeps ARP2/3 in an inactive state. **b** Phase II: Characterised by the recruitment of CDC42 and a sharp increase in ARF1 leading to the removal of PICK1. This allows for the activation of ARP2/3 and buildup of F-actin. CDC42 binds to the CRIB domain of IRSp53 thereby activating it. The SH3 domain of IRSp53 can now bind to ARP2/3 activators and create F-actin. **c** Phase III: Characterised by endocytic vesicle formation, the presence of CDC42, ARF1/GBF1 and F-actin

3000 (as per the manufacturer's instructions). Cells were subsequently processed for electron microscopy 24 h after transfection. Briefly, AGS cells were double transfected with GFP-tagged IRSp53 and an expression vector encoding for a GFP-binding peptide conjugated to APEX2 (Addgene plasmid #67651). Cells were washed in PBS, fixed in 2.5% glutaraldehyde in 0.1 M sodium cacodylate buffer and subjected to the 3,3′-diaminobenzidine (DAB; Sigma-Aldrich) reaction in the presence of $H_2O_2$ for 30 min. DAB reaction product was contrasted by 1% osmium tetroxide for 2 min. Cells were processed in situ and embedded in LX112 resin before sectioning parallel to the culture dish.

Tomogram: Thick plastic sections (200 nm) were cut on an ultramicrotome (UC6, Leica) and collected onto formvar-coated copper slot grids and lightly carbon coated. Dual-axis tilt series were acquired on a 120 kV TECNAI 12 (FEI) transmission electron microscope fitted with a LC-1100 4k x 4k lens coupled CCD camera (Direct Electron) and a tilt rotate holder (Fischione) utilising a tilt range of −60 to +60°. Microscope control and image acquisition was accomplished using the software SerialEM[68]. Tilt series were later reconstructed using weighted back projection and fiducial markers (10 nm) into a single volume with IMOD[69]. To examine areas with the greatest electron density in an unbiased manner, density-based thresholding was employed with the Isosurface render programme in IMOD as previously described[49] for APEX/DAB reaction product. The whole PM of the tomographic volume was rendered and tubular regions of interest were highlighted at greater magnification.

**Protein-retention expansion microscopy**. The ProExM protocol was adapted from previous reports[50,51]. IRSp53−/− GFP-IRSp53 addback cells were grown on coverslips for 2 days. The cells were fixed using 4% PFA for 15 min at room temperature (RT). The surface CD44 was stained using α-CD44 (Rat) following which the cells were permeablised using 0.05% Tween20 for 15 min. IRSp53 was dual stained using α-GFP and IRSp53 antibodies both of which were generated in Rabbit. Following this, secondary antibodies against Rat (Alexa-568) and Rabbit (Alexa-488) were used. The cells were then treated with Acryloyl X-SE (10 mg/ml stock solution in DMSO, used 1:100 diluted in PBS) for 12 h at RT. The cells were washed with PBS 2× 15 min each before proceeding to gelation. For a 10 ml of monomer solution (sod. acrylate (final concentration 8.6 g/100 ml), acrylamide (final concentration 2.5 g/100 ml), N,N′-methylenebisacrylamide (final concentration 0.15 g/100 ml), NaCl (final concentration 11.7 g/100 ml) were diluted in 1× PBS. Monomer solution (48 µl) was mixed with water (1 µl), TEMED (1 µl) and APS (10%, 1 µl) was added on to the cells. The cells were incubated at 37 °C for 30 min. The gel was incubated in the digestion buffer (50 mM Tris pH 8.0,1 mM EDTA, 0.5% Triton X-100, 0.8 M guanidine HCl, Proteinase K (1:100, final concentration 8 units/ml, added before use) for 6 h at 37 °C. The gel was washed with double distilled water for 3–5 times for 15 min to achieve full 4× expansion. The gel was placed on a coverslip and was imaged using ×100 spinning disk microscope.

**Image analysis**. In all cases, images were analysed with ImageJ and/or custom software written in MATLAB (The Mathworks, Natick, Massachusetts, USA). The number of cells and repeats of the experiments are mentioned in the legends and figures. Statistical significance ($p$) was calculated by Mann–Whitney $U$ test and two-sample Student's $t$ test, as reported in the legends.

pH pulsing assay analysis: A semi-automated analysis was developed in MATLAB to identify newly formed endocytic vesicles in the pH pulsing assay and trace their intensity over time in pH 7, pH 5 and RFP channels. The traces are an

average of many individual traces of all the endosomes pooled from different cells, which are compared with randomly placed spots within the cells. See S.I. for further details.

Endocytic assay analysis: In all cases, images were analysed with ImageJ and custom software written in MATLAB (The Mathworks, Natick, Massachusetts, USA). Each endocytic assay was performed with two technical duplicates. The number of repeats for each experiment is mentioned in its figure legend. For a given experiment, weighted mean for the technical duplicates was calculated as mentioned previously[14]. The total number of cells taken for analysis is mentioned in the legends and figures (at least 40–50 cells were taken from each technical duplicates). Subsequently, the cell-wise endocytosis distribution was normalised by weighted means of the control. This allowed data to be pooled from different days to be depicted as average and standard deviation as error bars. The statistical significance was calculated by Mann–Whitney U test. * represent p value <0.05.

Co-localisation analysis: The analysis was performed by two methods.

JaCoP (Just another co-loc plugin). An ImageJ plugin which has multiple options by which co-localisation between two molecules can be measured[70]. The object-based overlap or Van Steensel cross-correlation function options were used. Briefly, images were thresholded manually and were given appropriate parameters for either of the two options.

Spotter33. It is a custom MATLAB script (in-house) as described previously[11]. The algorithm consists of applying a top-hat filter on the images, following which the structures are segmented. The threshold is manually determined for each channel. The segmented structures are then trimmed (user determined) until their are shape and size matched to the original structures. In a channel, each segmented particle mask's centroid and pixel list is recorded, and the presence of a particle in that location is checked in the other channel. The number of pixels overlapped is normalied to the particle area. This value is averaged across all the particles for a given cell and reported.

Residence time analysis: FR-AGS cells expressing the desired molecule tagged with a fluorescent protein was imaged for the appropriate time 37 °C in TIRF. TagRFPt-PICK1 was imaged for 100 frames with 200 ms exposure and 500 ms interval and Nyquist criteria was satisfied. Spots were segmented and tracked using μ-track[71]. The residence time of TagRFPt-PICK1 spots per cell was calculated from the output of μ-track by custom MATLAB script. Briefly, the frame in which the spot was detected and the last frame a given spot of tracked is recorded. Spots that appeared in the first and the last frame of the movie are discarded. Additionally, spots which appeared for only 1 frame are also discarded. The spots whose track end was the last frame was discared as well as the track may or may not have continued.

**Data availability**. The authors declare that all data supporting the findings of this study are available within the paper and its Supplementary Information Files or from the authors on reasonable request.

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

## Acknowledgements

We thank Ramya Purkanti, Kabir Hussain and Balaji Ramalingam for help with the analysis, Neeraj Sebastian for making TagRFPt-CDC42, Rashmi Godbole for help with expansion microscopy, R.P. for her help with BAR domain database creation, Marcus Taylor (UCSF/NCBS) for mCherry-IRSp53 and SecGFP-TfR constructs, Gero Miesenböck (University of Oxford) for ecliptic-GPI, Paul Melançon (University of Alberta) for ARF1-mCherry, Catherine Jackson (National Institutes of Health) for GBF1-mCherry, Roland Wedlich-Soeldner (Universität Münster) for pRuby-lifeact, TagRFPt-PICK1 and GFP-PICK1 (Harvey McMohan, MRC), Mike Way (The Francis Crick Institute) for GFP-NWASPΔVCA and GFP-NWASP-VCA domain, J. Hanley (University of Bristol, UK) for pIRES-EGFP-PICK KK-EE mutant and N-WASP CA domain and Manoj Matthew from CIFF (Central Imaging and Flow cytometry Facility, NCBS) for helping to set up the pH pulsing setup. Authors were supported by Wellcome Trust-DBT India Alliance Early Career Fellowship (G.M.), NCBS-TIFR graduate student fellowship (M.S.). J.C. Bose Fellowship and a Margadarshi Fellowship (IA/M/15/1/502018) from the Wellcome Trust-DBT Alliance (S.M.), Wellcome Trust-DBT India Alliance Intermediate Fellowship and Simons Centre for Living Machines (M.T.), and grants, DBT-CoE (S.M), R.G.P was supported by the National Health and Medical Research Council (NHMRC) of Australia (programme grant, APP1037320 and Senior Principal Research Fellowship, 569452), and the Australian Research Council Centre of Excellence in Convergent Bio-Nanoscience and Technology (CE140100036), G.S. and A.D. were supported by Italian Association for Cancer Research Investigator Grant (10168 and 18621 to G.S.) and European Research Council (268836 to G.S.). We acknowledge the Australian Microscopy & Microanalysis Research Facility at the Centre for Microscopy and Microanalysis at The University of Queensland. We acknowledge the CIFF (Central Imaging and Flow cytometry Facility at National Centre for Biological Sciences, TIFR, India. We thank K. Joseph Matthew (S.M. Laboratory) for making the schematic in Fig. 1a and Fig. 8.

## Author contributions

M.S. and G.M. executed and analysed all fluorescence microscopy experiments. M.S. and M.T. developed the analysis. R.G.P. and J.R. performed, analysed and interpreted all electron microscopy experiments. A.D. and G.S. helped with the IRSp53 knockout and IRSp53 mutant addback cell line construction. G.M., M.S. and S.M. planned all experiments and wrote the manuscript with inputs from the remaining authors.
