## [Peer Review File · Nature Communications]

This manuscript has been previously reviewed at another journal that is not operating a transparent peer review scheme. This document only contains reviewer comments and rebuttal letters for versions considered at Nature Communications. Mentions of prior referee reports have been redacted.

Reviewers' comments:

Reviewer #1 (Remarks to the Author):

Review for manuscript by Sathe et al.

The manuscript by Sathe et al is quite impressive in terms of technology, as for the first time they have applied a pH sensitive vesicle formation assay (already used to characterize CCP formation by C. Merrifield) to the spatiotemporal characterization of some of the BAR domain proteins and small GTPases possibly involved in the formation of clathrin-independent vesicles, also known as CLICs. The CLIC/GEEC pathway has been the focus of intense investigation for the last few years and as such, this manuscript contributes to a better understanding of this still poorly characterized pathway. I believe however that additional experiments are warranted in order to better establish the role of the proteins studied here.

1. On page 5, it is mentioned that the assay detects two populations and that the authors discarded the population where *cdc42* and *secGFP-GPI* do not colocalize. It would be interesting to know if the GPI negative population corresponds to CLIC cargoes other than GPI-AP such as toxins (Shiga, Cholera) or CD44. This would substantiate the point raised in the discussion that "there is no unifying theme for vesicle scission in CIE and multiple modules may co-exist". They further argue that the discarded population may correspond to CME. This can be easily documented by blocking CME in these experiments.

2. Because the use of pH waves is a requisite for monitoring the precise kinetics of protein recruitment to CLICs in this assay, the authors ought to rule out potential artifacts, such as acidity, with other assays that can document the role of the tested proteins in CG formation. Moreover, while protein localization in time and space is interesting, it is not necessarily related to an active role in the pathway. Thus, the authors should show that down-modulating or knocking out the main candidates studied here (*Irsp53*, *PICK1*, *Cdc42*, *Arp2/3*) affects the uptake of known cargoes of the CLIC pathway (CD44, Shiga B-subunit...) which as mentioned in the introduction, has high endocytic capacity. Related to this point, endocytosis should be quantified at the level of the plasma membrane, as endosome labeling (see Fig 4) may not accurately reflect the uptake process per se.

3. It is unexpected that *Irsp53* is involved in membrane fission. As an I-BAR protein, it is rather expected to promote negative curvature and membrane protrusion rather than invagination. This effect is actually well shown by EM in figure S4. In figure 8, the authors conclude that *Irsp53* is localized on the edge of the invagination where curvature is positive. This assumption is not supported by the data. The authors should exclude possible mislocalization of the BAR domain protein due to its overexpression. Endogenous *Irsp53* should be better characterized.

4. The authors justify their selection of hits from Figure 2 based on interaction with *cdc42*. I therefore do not understand why they discard *GRAF1*. Several recent papers by the Lundmark lab have convincingly shown that *GRAF1* controls the CG pathway (see Cell Rep 2017 and JCS 2015) and pertinent to the present study, that membrane invagination is driven by transient interaction between *cdc42* and *GRAF1*.

Endophilin A is another key actor in CG vesicle formation, as shown by the work of Johannes and MacMahon (Ref 10 & 11). Endophilin is also involved in CG formation through actin organization (Ref 11). The authors did not select endophilin either, although like *GRAF1*, it was identified as a

hit in the RNAi screen (Fig 2). It is therefore important to know how these two BAR proteins are recruited to CG and how they fit in the model presented in Fig 8.

5. Several papers show that PICK1 is involved in CME and a recent JCB paper (Fiuza et al 2017) convincingly shows that PICK1 does so through binding to dynamin and AP2. How do the authors explain that PICK1 has no effect on Tf uptake in their study? This result questions the specificity of the PICK1 inhibitor used here. These studies should be included in the manuscript.

Minor:

Several experiments are performed only in duplicate whereas significant statistics usually require at least 3 independent experiments.

Reviewer #3 (Remarks to the Author):

Sathe et al. are exploring the molecular requirements for the CLIC/GEEC pathway, a form of non-clathrin dependent endocytosis that is independent of dynamin, caveolin, and endophilin. The authors use a method that employs rapid pH shifts, coupled with quantitative fluorescence microscopy, to measure the internalization of a model CLIC/GEEC cargo (SecGFP-GPI) that is predominantly internalized by this pathway. By following the association of various regulatory molecules with cargo, coupled with shRNA and drug studies, the authors propose a complex model driven by ARF1, its GEF (GBF1), the BAR-family protein PICK1 (a negative regulator of ARP2/3 function), CDC42, an additional BAR-family protein called IRSp53 (a positive regulator of ARP2/3), along with ARP2/3 and actin. Overall, the paper is interesting, and could present a significant advance in our understanding of the molecules that govern clathrin-independent pathways for endocytosis. However, some of the conclusions are not completely supported by the available data. I have the following comments/suggestions:

1. In their model, the PIs argue that IRSp53 is recruited by CDC42 to form an effector complex that promotes ARP2/3 activation by way of an unknown NPF. However, this conclusion is not well supported by the data. First, the recruitment of IRSp53 does not mimic the kinetics of CDC42 recruitment: IRSp53 shows maximal recruitment at around -25 s and a second wave at around -5 s (followed by an additional wave at +15 s), CDC42 appears to peak around +3 s. Second, ARP2/3 recruitment appears to begin very early on and peaks in the -15 s to -5 s time frame. Third, there are a very large fraction of SecGFP-GPI-positive vesicles (about 45%) that are not CDC42 positive. This large pool does not coincide with the fraction of SecGFP-GPI that is internalized by clathrin-dependent endocytosis. The authors need additional evidence (e.g., showing that IRSp53 recruitment depends on CDC42, and that ARP2/3 recruitment depends on IRSp53) to support this conclusion.

2. On line 250 and later in the Discussion (lines 394-396) the authors state that the loss of IRSp53 leads to a complete loss of CG endocytosis. However, this does not explain how the cell can still form CLICs (Fig. 3D), which are proposed to be GC-dependent carriers.

Minor points:

1. Lines 75-77 come out of the blue – there is no introduction of BAR-domain proteins, and why the reader would suppose that BAR domain proteins are involved.

2. Lines 148-152. With no mention of Fig. S2, it is difficult to understand what NoColc refers to.

3. Line 220 – affected should be affecting
4. Line 229. There is no Fig. S1G.
5. Line 233. We first, we counted.
6. Line 283. I don't understand how Fig. S4d relates to the preceding sentence.

Reviewer #4 (Remarks to the Author):

I am overall satisfied with the response. Yet, I continue to think that this is a technology-motivated work, and some of the conclusions may turn out to be incorrect, but I think the authors have done a major effort in addressing the criticism. I continue to dislike some of the conclusions. For instance "we find that IRSp53 co-localises with ARP2/3 at the TIRF plane (Figure S4d) indicating that IRSp53 does interact with ARP2/3". Co-localization and interaction are two very different things! It is also disappointing that while the Arp2/3 complex is implicated, the mechanism of Arp2/3 complex activation in this process (i.e. NPF) remains unresolved after the revisions. Yet, I would not like to ask again for things I already requested in the initial review.

Point by point response to Reviewer's comments

Reviewer #1 (Remarks to the Author):

Review of the manuscript by Sathe et al.

The manuscript by Sathe et al is quite impressive in terms of technology, as for the first time they have applied a pH-sensitive vesicle formation assay (already used to characterize CCP formation by C. Merrifield) to the spatiotemporal characterization of some of the BAR domain proteins and small GTPases possibly involved in the formation of clathrin-independent vesicles, also known as CLICs. The CLIC/GEEC pathway has been the focus of intense investigation for the last few years and as such, this manuscript contributes to a better understanding of this still poorly characterized pathway. I believe however that additional experiments are warranted in order to better establish the role of the proteins studied here.

We thank this reviewer for her/his encouraging remarks. Many of the points raised by this and the other reviewers had already been addressed in our "Point by Point Response to Reviewers", submitted along with the revised manuscript that was transferred to Nature Communications from Nature Cell Biology. In the interest of clarity, we now reproduce these responses as well as answer the remaining points as below:

1. On page 5, it is mentioned that the assay detects two populations and that the authors discarded the population where cdc42 and secGFP-GPI do not colocalize. It would be interesting to know if the GPI negative population corresponds to CLIC cargoes other than GPI-AP such as toxins (Shiga, Cholera) or CD44. This would substantiate the point raised in the discussion that "there is no unifying theme for vesicle scission in CIE and multiple modules may co-exist".

Response: There are several points raised here that need to be addressed individually.

In our pH pulsing assay, a pH5 protected secGFP-GPI spot can arise in the TIRF plane from the following sources: 1) an endocytic vesicle formed at the boundary of pH7 – 5 switch; 2) a secretory vesicle with luminal pH 7; 3) a recycling vesicle with luminal pH 7 and 4) a vesicle (of unknown origin) coming into our window of observation by lateral movement from elsewhere else in the cell. Therefore, in our study as well as that of the Merrifield group^{1,2}, several conditions were imposed to remove as many false positives as possible (using automated and manual checks). This resulted in discarding a large proportion of secGFP-GPI spots. Unlike in the case of CME, where the cargo secGFP-TfR is measurably concentrated at the time of endocytic vesicle formation (Supplementary Fig. 2c), the secGFP-GPI spots do not show a prior accumulation/concentration in the pH 7 channel (Fig. 1b-c). Thus, we imposed an additional criterion to select the secGFP-GPI endocytic events, namely those that co-occurred along with the CG pathway molecules during the period under observation (-18 to +18 sec). This divides the identified population of vesicles into two. In all instances, we find that the CG endocytic molecular component localizes to a majority (>55-75%) of the endocytic spots identified in such a manner.

In our study, we focus only on the subset of GPI-AP endosomes that co-localise with either CDC42 or other molecules players that have been shown to perturb the CG uptake. It should be pointed out that under steady-state conditions, a minority (~ 20-30%) does co-occur with dynamin/clathrin as expected from our previous work³. However, as we show in Supplementary Fig. 2d-e the profile

of recruitment of dynamin and clathrin to even these minority endocytic vesicles is not typical (see Supplementary Fig. 2c and Ref. 3, 4). By contrast, when we imaged the secGFP-TfR along with dynamin and clathrin, >80% of the bonafide endocytic spots of secGFP-TfR recruited dynamin consistent with previous work in the field (Supplementary Fig. 2c & Refs. 3, 4).

At present, we cannot comment on the nature of GPI-AP negative population of CG endocytic vesicles since our assay relies solely on the detection of pH5-protected SecGFP-GPI spots. The population where GPI-AP and CDC42 do not co-localize is a set of *GPI-AP-containing* endocytic vesicles that *do not* detectably recruit CDC42. The technical reasons behind lack of detection of CDC42 are detailed in the lines 162-166 of the revised manuscript and lines 127-199 of the supplementary methods. Briefly, these are due to the background signal from auto-fluorescence contributing to the low signal to noise at the levels of expression that we have used for our experiments.

Nevertheless, this reviewer raises an interesting point whether there are many mechanisms of CG endocytosis. The study of these populations of endocytic events, while of great interest, is beyond the scope of the present work, since it would require an entirely new set of experiments, including the development of 3-colour TIRF imaging with two pH sensitive tags with the third colour being a tag for the alternate pathway. This is challenging and would require new instrumentation and probes at our end, and beyond the scope of this study.

We have shown earlier that, CLICs when purified biochemically show the presence of CD44 and Thy-1 (GPI-AP) ⁴. In migrating cells, CLICs are enriched at the leading edge. These polarized CLICs are populated by CD44, Thy-1 and GFP-GPI. All these endosomes showed co-localisation with Cholera toxin ⁴, indicating that there may be a single class of endocytic vesicles. However, like the clathrin-derived vesicles, it is likely that there is a complex interplay between cargo and endocytic process, resulting in a heterogeneity of CG endocytic vesicles.

“They further argue that the discarded population may correspond to CME. This can be easily documented by blocking CME in these experiments.

As suggested by this reviewer we attempted blocking CME entry routes through pharmacological or siRNA approaches. This has led to technical difficulties in doing repeated buffers exchanges in situ since the adhesive properties of the cells change. However, blocking the dynamin-endocytic route we do find that endocytosis of GPI-APs and the fluid phase is relatively unaffected ^{3,5,6}, as it is when all isoforms of dynamin are deleted ⁶.

2. Because the use of pH waves is a requisite for monitoring the precise kinetics of protein recruitment to CLICs in this assay, the authors ought to rule out potential artefacts, such as acidity, with other assays that can document the role of the tested proteins in CG formation. Moreover, while protein localization in time and space is interesting, it is not necessarily related to an active role in the pathway. Thus, the authors should show that down-modulating or knocking out the main candidates studied here (Irsp53, PICK1, Cdc42, Arp2/3) affects the uptake of known cargoes of the CLIC pathway (CD44, Shiga B-subunit...) which as mentioned in the introduction, has the high endocytic capacity. Related to this point, endocytosis should be quantified at the level of the plasma membrane, as endosome labelling (see Fig 4) may not accurately reflect the uptake process per se.

1. The reviewer raises an important point that changing the pH of the outside medium might affect endocytosis and must be ruled out as a potential artefact. The longest pH pulsing movie generated during our experiments was for 10 minutes. Keeping that in mind, the longest a cell might face pH 5 is around 5 minutes. Thus, we pre-treated the cells with pH 5.5 buffer for 5 minutes followed by a 5-minute fluid pulse. We find that pre-exposure to pH 5.5 buffer does not affect fluid-phase endocytosis appreciably. We have added this in the revised version of the manuscript (Supplementary Fig.1e) and clarified this in the text (see lines 112-115 in the revised manuscript).

2. We agree with the reviewer's point that recruitment observed solely by using pH pulsing is insufficient as a criterion to assess if a component is involved in the machinery for the endocytic event. Thus, all the molecules tested in our manuscript were independently verified as affecting the CG endocytic pathway and not CME (except for ARP2/3) using alternate methods in multiple cell lines (mammalian and fly) described in both our previous work and the current manuscript.

We list these alternate experiments below:

a. CDC42: dominant negative mutants, pharmacological inhibitors ^{3,7},

b. ARF1: knockdown and dominant negative mutants ⁸,

c. GBF1 knockdown, dominant negative mutant over-expression ⁹ and by using pharmacological inhibitor in the revised manuscript (Supplementary Fig.1f-g, Fig.3c-d),

d. F-actin: pharmacological inhibitors ⁷,

e. ARP2/3: is tested in our earlier work by knockdown ¹⁰ and by using pharmacological inhibitor in the revised manuscript (Fig.6a), also affects CME ¹¹ (Fig.6a).

f. PICK1: tested by knockdown and pharmacological inhibitors in the revised manuscript (Fig.7b & 7c),

g. IRSp53: tested in a knockout cell line along with addback with specific deletions/mutations of the IRSp53 gene (Fig.3 & 4).

3. We agree with the reviewer that Figure 4a relates to counting the number of short pulse endosomes per cell. We have done additional experiments with the necessary normalization asked by the reviewer. We urge the reviewer to see Figure 3b-d and Figure 4d, wherein we have quantified average endocytosis per cell. Here, the total amount of endocytosed material is normalised to the projected cell area.

3. It is unexpected that Irsp53 is involved in membrane fission. As an I-BAR protein, it is rather expected to promote negative curvature and membrane protrusion rather than invagination. This effect is actually well shown by EM in figure S4. In figure 8, the authors conclude that Irsp53 is localized on the edge of the invagination where the curvature is positive. This assumption is not supported by the data. The authors should exclude possible mislocalization of the BAR domain protein due to its overexpression. Endogenous Irsp53 should be better characterized.

We agree with the reviewer that endosome formation requires membrane invagination instead of the protrusion. However, the neck of a vesicle may provide the necessary negative curvature required for IRSp53 recruitment (see Reviewer Figure 1a). In Fig.8, we refer to regions wherein curvature is indeed negative (see blue regions of the schematic).

Assessing endogenous IRSp53 localization has been a challenge so far, due to the lack of an immunoEM compatible antibody (we have tried antibodies on ultrathin frozen sections of WT and KO MEFs with no success; a future aim is the development of new antibodies). Thus, we chose to generate thick section tomograms of the cells co-expressing GFP-IRSp53 and APEX. GBP-APEX binds to GFP and converts 3,3'-diamino-benzamide into an osmiophilic polymer in presence of H₂O₂¹². This technique relies on low levels of expression of GFP-IRSp53. High expression leads to excessive cytosolic expression of IRSp53 (which is not the case in the EM experiments in which we see a very specific localization to membranes). Most importantly, in our EMs as an internal positive control, we show, that IRSp53 localises to filopodia tips (Supplementary Fig. S4c) as previously reported¹³. We have also shown in our previous work that the cellular localization of GFP-IRSp53 and endogenous IRSp53 show overlap^{14,15} particularly when the levels of GFP-IRSp53 expression is kept low (Reviewer Figure 1b).

Further, in the revised manuscript we have included our results from protein retention expansion microscopy or ProExM^{16,17}. ProExM works by physically separating the fluorescent proteins by anchoring them in a swellable gel. We performed proExM up to 4x in IRSp53^{-/-} cell re-expressing GFP-IRSp53 at near endogenous levels were imaged in the gel at 100X in a spinning disk confocal microscope (Supplementary Fig.4e). The cells were immunostained for antibodies against CD44 and IRSp53. We find several instances of CD44 marked invaginations at various stages of progression that show recruitment of IRSp53. At stages, wherein the neck is visibly constricted we see indications of the enrichment of IRSp53 at the neck relative rest of the invagination. Similar to our observations with EM, we see IRSp53 being enriched at the tips of the filopodia. This information is now available as Supplementary Fig. 4e.

4. The authors justify their selection of hits from Figure 2 based on interaction with cdc42. I, therefore, do not understand why they discard GRAF1. Several recent papers by the Lundmark lab have convincingly shown that GRAF1 controls the CG pathway (see Cell Rep 2017 and JCS 2015) and pertinent to the present study, that membrane invagination is driven by the transient interaction between cdc42 and GRAF1.

Endophilin A is another key actor in CG vesicle formation, as shown by the work of Johannes and MacMahon (Ref 10 & 11). Endophilin is also involved in CG formation through actin organization (Ref 11). The authors did not select endophilin either, although like GRAF1, it was identified as a hit in the RNAi screen (Fig 2). It is therefore important to know how these two BAR proteins are recruited to CG and how they fit in the model presented in Fig 8.

a) We would like to clarify that the rationale behind the selection of hits. CG32082 and MIM, the hits from Fig.2a-b were interesting as they possessed an I-BAR domain. Among the mammalian orthologues¹⁸ of CG32082 and MIM, IRSp53 is the only protein that has a CRIB domain¹⁹ and is the most studied. Therefore, we chose to focus on IRSp53 as our most likely candidate to play a major role in the process. We will in our future studies like to explore the role of other I-BAR domain family members.

b) Regarding GRAF1, we did look at the recruitment of GRAF1 using the pH pulsing assay and we found that a significant accumulation of GRAF1 begins at -10s coincident with the recruitment of CDC42 and remained in the endocytic sites even after pinching (till around +12s) (Reviewer Figure 1c). This supports the notion that GRAF1 can stabilize tubules that would start to appear proximal to pinching and then persist while the CLICs are in the vicinity of the membrane. Thus, it is likely to be required for a later stage of the internalization process, such as driving the

endocytic vesicles towards fusion with later compartments and preventing their regurgitation. Here, we did not focus on GRAF1 at this stage, because we believe that it is required after the vesicle formation. However, it is clear that GRAF is involved in the process and is a hit in the limited screen (see line 204-205 in the revised manuscript).

c) The focus of the manuscript is to identify molecular machinery that is required for vesicle formation and at present IRSp53 and PICK1 are better candidates. Firstly, they are recruited only during vesicle formation and fall off at or before the scission (Fig.5a and 7f). Secondly, removal of IRSp53 abolishes CG endocytosis as documented in the IRSp53^{-/-} lines (Fig.3d & 4a-c).

d) Endophilin A: We did not pursue a role of Endophilin A in CG pathway because we do not think it is involved in the CDC42-dependent CG endocytosis. Instead, it is part of an alternate CIE pathway which is likely dynamin-dependent²⁰⁻²².

5. Several papers show that PICK1 is involved in CME and a recent JCB paper (Fiuza et al 2017) convincingly shows that PICK1 does so through binding to dynamin and AP2. How do the authors explain that PICK1 has no effect on Tf uptake in their study? This result questions the specificity of the PICK1 inhibitor used here. These studies should be included in the manuscript.

Hanley's group has characterized the role of PICK1 in AMPA receptor internalization upon NMDA stimulation in neuronal cells extensively over the years. They have reported that PICK1 inhibits ARP2/3 based actin polymerization^{23,24}. PICK1 also connects CDC42 to AMPA receptor internalization²⁵. The latest work that this reviewer has referred to from the same group suggests that PICK1 can additionally bind to AP2 and dynamin as well²⁶. However, they also show that depletion of PICK1 does not affect Tf uptake similar to our observations (Fig. 7c, Ref. 26). Fiuza *et al* 2017 show that interaction of PICK1 with dynamin occurs via its BAR domain. Clearly, AMPA receptor internalization is very interesting and complex but we cannot comment on the precise nature of the endocytic mechanism at play in that case. Thus, PICK1, on one hand, in neuronal cells regulates the internalization of AMPA receptor via a clathrin and dynamin-dependent pathway. On the other hand, it does not affect the typical cargo of clathrin-mediated endocytosis, the Transferrin Receptor, consistent with our results shown (Fig.7b-d).

Minor:

Several experiments are performed only in duplicate whereas significant statistics usually require at least 3 independent experiments.

As detailed in our supplementary information section, we estimate statistical significance by comparing the cell-wise distribution pooled from multiple experiments. We consider each cell when we are assessing the statistical significance of results. However, we think reviewer's issue might be due to the way we have chosen to represent data. Instead of showing a representative bar graph, we prefer to show the data from all the experiments. Each endocytic assay was performed with 2 technical duplicates. The number of repeats of each experiment is mentioned in its figure legend. For a given experiment, weighted mean for the technical duplicates was calculated as mentioned previously⁸. The total number of cells taken for analysis is mentioned in the legends and figures (at least 40-50 cells were taken from each technical duplicates). Subsequently, the cell-

wise endocytosis distribution was normalised by weighted means of the control. This allowed data to be pooled from different days to be depicted as average and standard deviation with error bars.

Reviewer #3 (Remarks to the Author): Sathe et al. are exploring the molecular requirements for the CLIC/GEEC pathway, a form of non-clathrin dependent endocytosis that is independent of dynamin, caveolin, and endophilin. The authors use a method that employs rapid pH shifts, coupled with quantitative fluorescence microscopy, to measure the internalization of a model CLIC/GEEC cargo (SecGFP-GPI) that is predominantly internalized by this pathway. By following the association of various regulatory molecules with cargo, coupled with shRNA and drug studies, the authors propose a complex model driven by ARF1, its GEF (GBF1), the BAR-family protein PICK1 (a negative regulator of ARP2/3 function), CDC42, an additional BAR-family protein called IRSp53 (a positive regulator of ARP2/3), along with ARP2/3 and actin. Overall, the paper is interesting and could present a significant advance in our understanding of the molecules that govern clathrin-independent pathways for endocytosis. However, some of the conclusions are not completely supported by the available data. I have the following comments/suggestions:

We thank this reviewer for her/his positive comments.

1. In their model, the PIs argue that IRSp53 is recruited by CDC42 to form an effector complex that promotes ARP2/3 activation by way of an unknown NPF. However, this conclusion is not well supported by the data. First, the recruitment of IRSp53 does not mimic the kinetics of CDC42 recruitment: IRSp53 shows maximal recruitment at around -25 s and the second wave at around -5 s (followed by an additional wave at +15 s), CDC42 appears to peak around +3 s. Second, ARP2/3 recruitment appears to begin very early on and peaks in the -15 s to -5 s time frame. Third, there is a very large fraction of SecGFP-GPI-positive vesicles (about 45%) that are not CDC42 positive. This large pool does not coincide with the fraction of SecGFP-GPI that is internalized by clathrin-dependent endocytosis. The authors need additional evidence (e.g., showing that IRSp53 recruitment depends on CDC42 and that ARP2/3 recruitment depends on IRSp53) to support this conclusion.

There are multiple points raised here and these are discussed separately.

1. The pH pulsing assay detects two populations of SecGFP-GPI spots, one positive for a given X-RFP molecule and another negative. For instance, the fraction of SecGFP-GPI that showed co-localisation with CDC42 [CDC42 Coloc] exhibited a profile similar to [CDC42 All] population (compare, Fig.1d & Supplementary Fig. 2a). The other fraction (CDC42 NoColoc) failed to show a discernable accumulation and its profile was similar to Random (Supplementary Fig.2b). Thus, we can only speculate on the nature of SecGFP-GPI spots that are CDC42 negative [CDC42 NoColoc]. While the reasons for not detecting CDC42 at all endocytic events is a function of both the signal and noise in the data, it may also reflect a genuine lack of recruitment at a given endocytic event. In addition, when we imaged recruitment of clathrin or dynamin along with SecGFP-GPI, we observed that around 30% of the traces showed some kind of recruitment, different from the random (Supplementary Fig.2d-e).

Thus, a combination of lack of genuine recruitment of CDC42 and a small fraction of SecGFP-GPI trafficking via CME, and the signal to noise of CDC42 at a given SecGFP-GPI endocytic site could be potential reasons why we failed to detect CDC42 in the entire population. The fraction of X-RFP NoColc varies among different molecules that we have tested in our manuscript (see Table

2). Thus, the only point that we can make conservatively is that molecules belonging to CG pathway and CME show association with SecGFP-GPI roughly in 70-30 ratio respectively.

This is detailed in the supplementary information, method section (Lines 127-199).

Regarding the way we determine an XFP-negative pool, please also refer to the explanation provided to Point 1, Reviewer 1 above.

2. We do not suggest that CDC42 is recruiting IRSp53, and thank this reviewer for pointing out this confusion. IRSp53, as the reviewer rightly points out is recruited quite early on. Based on our previous work¹⁴ we suggest that CDC42 recruitment leads to the activation of IRSp53, allowing the latter to bind to its effectors. This mode of action has been previously observed in the case of filopodia formation. IRSp53 through its I-BAR, PIP2-binding domain is enriched at the plasma membrane but held in a closed conformation that can be open up following binding of CDC42 to its CRIB domain. It is also of note that in the closed conformation, some of the authors of this work showed that IRSp53 also display weak, but detectable capping activity, which is relieved following CDC42 binding^{14,19}. The recruitment of additional NPF would then be needed for the activation of ARP2/3.

3. ARP2/3 recruitment occurs prior to CDC42 arrival, likely to be mediated by PICK1. PICK1 recruits ARP2/3 and keeps it in an inactive state. This interpretation is supported by the poor signal of F-actin filaments during the peak recruitment of PICK1. ARP2/3 activation, inferred by a rise in F-actin accumulation, occurs coincidental to CDC42 recruitment. Moreover, addback in the IRSp53^{-/-} MEFs, of an IRSp53 mutant incapable of binding to CDC42, failed to rescue CG endocytosis. This strengthens the interpretation that IRSp53::CDC42 interaction is required for CG endocytosis.

2. On line 250 and later in the Discussion (lines 394-396) the authors state that the loss of IRSp53 leads to a complete loss of CG endocytosis. However, this does not explain how the cell can still form CLICs (Fig. 3D), which are proposed to be GC-dependent carriers.

The conclusion that a loss of IRSp53 leads to a complete loss of CG endocytosis is based on multiple results:

1. In the IRSp53^{-/-} MEFs, the fluid-phase endocytosis becomes insensitive to GBF1 inhibition (by LG186) (Fig. 3c), while upon re-introduction of IRSp53WT in the IRSp53^{-/-} MEFs, the fluid-phase uptake is rendered sensitive to GBF1 inhibition similar to that observed in the WT MEFs.

2. In Fig.3d, GBF1 inhibition (by LG186) and loss of IRSp53 both lead to a reduction in structures that are classified as CLICs using HRP as a general endocytosis tracer. HRP marks all the endocytic structures originating from the plasma membrane regardless of the specific endocytic pathway used. We lose CLICs as determined by EM. In the absence of specific markers for the CG pathway, we always need to use strict morphological criteria but in thin sections, there is always some morphological overlap between the 2 types of structures [e.g. tubular elements of EE (early endosomes) vs. CLICs, even at 2 minutes]. Thus, the CLIC number does not fall to zero during IRSp53 removal and GBF1 inhibition.

The classification and quantitation are done as follow: 5-minute HRP uptake images were captured at random across the monolayer by moving a defined distance across the grid to avoid user bias.

For 2-minute uptake, whole cells positive for HRP were imaged and montaged to generate a high-resolution image encompassing the entire cell. 5 cells were imaged and montaged for each replicate. HRP-labelled elements per image or per cell profile were classified using the following criteria; vesicular elements including clathrin-coated vesicles and caveolae – circular profiles <200 nm in diameter; early endosomes – circular and ring-shaped profiles, including multivesicular structures >200 nm in diameter; CLIC/GEEC - other profiles including tubules and small ring-shaped structures <200 nm in diameter.

3. In accordance with the above-mentioned data, we further observe that in the absence of IRSp53, CG cargoes such as fluid-phase and GPI-AP traffic via CME derived vesicles, since they appear more colocalized with endocytosed TfR.

These results together lead us to conclude that loss of IRSp53 results in a nearly complete loss of CG endocytosis.

Minor points:

1. Lines 75-77 come out of the blue – there is no introduction of BAR-domain proteins, and why the reader would suppose that BAR domain proteins are involved.

As mentioned in Lines 75-77, the lack of any discernable coat, lead naturally to other curvature sensing proteins such as BAR domain. BDPs are well-known curvature sensing proteins in membrane trafficking and we have referred to the relevant literature.

2. Lines 148-152. With no mention of Fig. S2, it is difficult to understand what NoColc refers to.

We thank the reviewer for pointing this out. We have now referred to Supplementary Fig.2 extensively along with supplementary methods in the lines highlighted. Supplementary Fig.1a is referred in Line 145, while Supplementary Fig.2b is referred in line 152. Additionally, we apologize for a typo in line 157 wherein we erroneously called S1a instead of S2a that has been remedied.

3. Line 220 – affected should be affecting.

We thank the reviewer for pointing this out. We have remedied this mistake.

4. Line 229. There is no Fig. S1G.

We apologize for incorrect figure calling. It is S1e instead of S1g. We have remedied this mistake.

5. Line 233. We first, we counted.

We have remedied this mistake.

6. Line 283. I don't understand how Fig. S4d relates to the preceding sentence.

We apologize for incorrect figure calling. We have remedied this mistake.

Reviewer #4 (Remarks to the Author):

I am overall satisfied with the response. Yet, I continue to think that this is a technology-motivated work, and some of the conclusions may turn out to be incorrect, but I think the authors have done a major effort in addressing the criticism. I continue to dislike some of

the conclusions. For instance "we find that IRSp53 co-localises with ARP2/3 at the TIRF plane (Figure S4d) indicating that IRSp53 does interact with ARP2/3". Co-localization and interaction are two very different things! It is also disappointing that while the Arp2/3 complex is implicated, the mechanism of Arp2/3 complex activation in this process (i.e. NPF) remains unresolved after the revisions. Yet, I would not like to ask again for things I already requested in the initial review.

We thank this reviewer for her/his agreement with points we have chosen to address. We show that IRSp53 localize to ARP2/3 puncta at the cell membrane (Supplementary Fig.4d), and also at the time of endocytic uptake (Fig.5a & 6b for IRSp53 and ARP3 respectively). However, as we had indicated earlier (and reiterate below) the battery of possibilities of the linker between IRSp53 and ARP2/3 is large. Here we leave this as a major open question for future research. Indeed, preliminary experiments in the laboratory indicate that VASP but not EPS8 might be implicated; however, we believe that these data will be the subject of a separate manuscript.

References:

1. Howes, M. T. et al. Clathrin-independent carriers form a high capacity endocytic sorting system at the leading edge of migrating cells. *J. Cell Biol.* 190, 675–691 (2010).
2. Sabharanjak, S., Sharma, P., Parton, R. G. & Mayor, S. GPI-anchored proteins are delivered to recycling endosomes via a distinct cdc42-regulated clathrin-independent pinocytic pathway. *Dev. Cell* 2, 411–423 (2002).
3. Merrifield, C. J., Perrais, D. & Zenisek, D. Coupling between clathrin-coated-pit invagination, cortactin recruitment, and membrane scission observed in live cells. *Cell* 121, 593–606 (2005).
4. Taylor, M. J., Perrais, D. & Merrifield, C. J. A high precision survey of the molecular dynamics of mammalian clathrin-mediated endocytosis. *PLoS Biol.* 9, e1000604 (2011).
5. Chadda, R. et al. Cholesterol-sensitive Cdc42 activation regulates actin polymerization for endocytosis via the GEEC pathway. *Traffic* 8, 702–717 (2007).
6. Kumari, S. & Mayor, S. ARF1 is directly involved in dynamin-independent endocytosis. *Nat. Cell Biol.* 10, 30–41 (2008).
7. Gupta, G. D. et al. Population distribution analyses reveal a hierarchy of molecular players underlying parallel endocytic pathways. *PLoS One* 9, (2014).
8. Gupta, G. D. et al. Analysis of endocytic pathways in *Drosophila* cells reveals a conserved role for GBF1 in internalization via GEECs. *PLoS One* 4, (2009).
9. Ariotti, N. et al. Modular Detection of GFP-Labeled Proteins for Rapid Screening by Electron Microscopy in Cells and Organisms. *Dev. Cell* 35, 513–525 (2015).
10. Disanza, A. et al. CDC42 switches IRSp53 from inhibition of actin growth to elongation by clustering of VASP. *EMBO J.* 32, 2735–50 (2013).
11. Disanza, A. et al. Regulation of cell shape by Cdc42 is mediated by the synergic actin-bundling activity of the Eps8 – IRSp53 complex. *J. Cell Biol.* 174, 101–112 (2006).

12. Chen, F., Tillberg, P. W. & Boyden, E. S. Expansion microscopy. *Science* (80-.). 347, 543–548 (2015).
13. Tillberg, P. W. et al. Protein-retention expansion microscopy of cells and tissues labeled using standard fluorescent proteins and antibodies. *Nat. Biotechnol.* 34, 987–992 (2016).
14. Scita, G., Confalonieri, S., Lappalainen, P. & Suetsugu, S. IRSp53: crossing the road of membrane and actin dynamics in the formation of membrane protrusions. *Trends Cell Biol.* 18, 52–60 (2008).
15. Kast, D. J. et al. Mechanism of IRSp53 inhibition and combinatorial activation by Cdc42 and downstream effectors. *Nat. Struct. Mol. Biol.* 21, 413–22 (2014).
16. Boucrot, E. et al. Endophilin marks and controls a clathrin-independent endocytic pathway. *Nature* 517, 460–5 (2015).
17. Renard, H.-F. et al. Endophilin-A2 functions in membrane scission in clathrin-independent endocytosis. *Nature* 517, 493–6 (2015).
18. Simunovic, M. et al. Friction Mediates Scission of Tubular Membranes Scaffolded by BAR Proteins. *Cell* 170, 172–184.e11 (2017).
19. Rocca, D. L., Martin, S., Jenkins, E. L. & Hanley, J. G. Inhibition of Arp2/3-mediated actin polymerization by PICK1 regulates neuronal morphology and AMPA receptor endocytosis. *Nat. Cell Biol.* 10, 259–71 (2008).
20. Rocca, D. et al. The Small GTPase Arf1 Modulates Arp2/3-Mediated Actin Polymerization via PICK1 to Regulate Synaptic Plasticity. *Neuron* 79, 293–307 (2013).
21. Rocca, D. L. & Hanley, J. G. PICK1 links AMPA receptor stimulation to Cdc42. *Neurosci. Lett.* 585, 155–159 (2015).
22. Fiuza, M. et al. PICK1 regulates AMPA receptor endocytosis via direct interactions with AP2 α -appendage and dynamin. *J. Cell Biol.* 216, 3323–3338 (2017).

a

b

c

Reviewer Figure 1: (a) Schematic representing different curvatures that exist in a budding endocytic vesicle. (b) Mouse keratinocytes showing that endogenous IRSp53 (stained using our mouse mAB antibody) and ectopically expressed GFP-IRSp53. (c) Recruitment profile of mCherry- GRAF1 to a forming SecGFP-GPI endocytic site. (n= 79 SecGFP spots).

REVIEWERS' COMMENTS:

Reviewer #1 (Remarks to the Author):

The experiments that I suggested aimed at better establishing the role of IRSp53 in CIE by confirming the data with other CG-dependent cargoes (CD44, Shiga toxin...) and by integrating the role of IRSp53 with known CIE actors such as GRAF1 and endophilin. These experiments have not been performed in the revised manuscript.

I have no doubts that IRSp53 is involved in CG endocytosis. Nevertheless, more substantial data are required to rule out possible biases induced by IRSp53 overexpression and to establish the unexpected role of an I-BAR protein in inward budding. Although I understand that it is technically challenging to localize endogenous Irsp53 by immuno EM, the new data presented in Fig S4e do not support the localization claimed by the authors. Since this localization is essential to validate the function of Irsp53 in the CG pathway, the model presented on Fig 8 is not supported by experimental data.

Reviewer #3 (Remarks to the Author):

The authors have addressed my concerns.

On behalf of my colleagues, I thank you for the opportunity to respond to the comments of Reviewer 1, reproduced below:

‘The experiments that I suggested aimed at better establishing the role of IRSp53 in CIE by confirming the data with other CG-dependent cargoes (CD44, Shiga toxin...) and by integrating the role of IRSp53 with known CIE actors such as GRAF1 and endophilin. These experiments have not been performed in the revised manuscript.

I have no doubts that IRSp53 is involved in CG endocytosis. Nevertheless, more substantial data are required to rule out possible biases induced by IRSp53 overexpression and to establish the unexpected role of an I-BAR protein in inward budding. Although I understand that it is technically challenging to localize endogenous Irsp53 by immuno EM, the new data presented in Fig S4e do not support the localization claimed by the authors. Since this localization is essential to validate the function of Irsp53 in the CG pathway, the model presented on Fig 8 is not supported by experimental data. ‘

We emphasize that despite the presence of multiple endocytic pathways only Clathrin-Mediated Endocytosis (CME) remains a relatively well-characterised while, the CLIC/GEEC (CG) pathway identified over a decade ago, remains poorly understood. By using a combination of electron microscopy and real-time live microscopy as well as molecular genetic and chemical perturbations, we identify two new BAR domain-containing proteins that provide an operational basis for the functioning of this pathway. The results obtained here are of broad relevance as they attempt to characterise how a cell can form a vesicle in absence of a coat clathrin and dynamin – the canonical pair of molecules required for conventional endocytosis.

Please find our response point-by-point as below:

1. We must emphasize that CG endocytosis is a subset of clathrin-independent pathways (CIE) and should not be used interchangeably. There are several CIEs (Mayor & Pagano, 2007; Mayor, Parton, & Donaldson, 2014) and all CIEs are by definition clathrin-independent. The CG pathway is characterized by being both clathrin and dynamin-independent. This is supported by data shown by our laboratory and now several others, in addition to what we present here. The endophilin-dependent pathway is another CIE that is clathrin-independent, but is dynamin-dependent (Boucrot et al., 2015; Renard et al., 2015).
2. IRSp53 knockout cells display almost complete loss of CLICs (Figure 3d). The residual fluid-phase endocytosis in the IRSp53 null cells is insensitive to LG186, a small molecule inhibitor against GBF1 (Figure 3c; GBF1 is one of the earliest known molecules that initiates CG endocytosis). The residual fluid and GPI-AP containing endosomes traffic via clathrin-mediated endocytic route in the IRSp53 null cells (Figure 4b). Furthermore, reintroduction of IRSp53 at endogenous levels into null cells (not over-expression as mis understood by Reviewer 1), restores the LG186-sensitive endocytosis to levels similar to the wild type (Figure 3b, c). This system also allows us to test various IRSp53 mutants to assess different requirements for its function in CG endocytosis (Figure 4d).
3. Testing the role of IRSp53 in internalization of different cargoes: We have looked at specific CG cargo, GPI-anchored protein and found its internalization to be greatly

reduced in IRSp53 knockout cells (Figure 4a). The residual GPI-anchored proteins containing endosomes in IRSp53 knockout actually traffic via the Clathrin-mediated endocytosis (Figure 4b), since they co-localize with the Transferrin receptor. At this time we are examining the routes that CD44 take in a separate manuscript that deals with CD44 organization and its trafficking. Shiga toxin, as suggested by the reviewer is not an exclusive CG cargo. Instead, it is also cargo for the endophilin-dependent endocytic pathway and like Cholera Toxin may enter via many endocytic modes (Renard et al, 2015).

4. Integrating the role of IRSp53 with endophilin in CG endocytosis: We did not pursue a role of Endophilin A in CG pathway because as mentioned in point 2, endophilin-dependent endocytosis has been shown to require dynamin function; it is not CG endocytosis. Further, dsRNA screen against BAR domain proteins as reported in our manuscript (Figure 2) shows that knockdown of Endophilin does not affect CG endocytosis in fly cells. Moreover, Boucrot et al., 2015 shows that endophilinA knockdown does not affect GPI-AP uptake in mammalian cells.
5. Integrating role of IRSp53 with GRAF1 in CG endocytosis: According to our screen and a published report from one of us (Lundmark et al., 2008), GRAF1 is indeed involved in CG endocytosis. GRAF1 is recruited to endosomes containing CG cargo and stays on until at least 2 minutes post internalization (Lundmark et al., 2008). Hence, when we examined the recruitment of GRAF1 using the pH pulsing assay we found that a significant accumulation of GRAF1 begins at -10s coincident with the recruitment of CDC42 and remained in the endocytic sites even after pinching (till around +12s) (Reviewer Figure 1c). This supports the notion that GRAF1 can stabilize tubules that would start to appear proximal to pinching and then persist while the CLICs are in the vicinity of the membrane. Thus, it is likely to be required for a later stage of the internalization process, such as driving the endocytic vesicles towards fusion with later compartments and preventing their regurgitation. Therefore, we did not focus on GRAF1 at this stage, because we believe that it may required after the vesicle formation step. These are directions that we are pursuing in ongoing work, but this is out of the scope of this current manuscript that is focused on the early events in vesicle formation in the CG pathway.

6. IRSp53 localization: We are happy to soften our conclusions about the precise localization by making a statement that summarises our observations. However, we would like to point out that all of the localization experiments, (however unconvincing to the Reviewer1) always had an internal positive control i.e. filopodia tip localization. Any additional EM experiments would require new antibodies. We presented evidence from EM in our first submission to NCB (Supplementary Figure 4c in the current version). The precise IRSp53 localization appears just beyond current experimental capabilities. In response to the concern raised by the Reviewers we have tried i) 3-D tomography using the APEX-EM technique (Figure 5c), ii) immunoEM with available antibodies (unsuccessful), iii) STED imaging (unsuccessful due to technical reasons) and iv) finally super-resolution imaging using Protein-retention Expansion microscopy (Supplementary Figure 4e). We believe that given available methodology we have tried our utmost to localize IRSp53 precisely. Therefore, we believe that this demand fits with the comment - this is an 'unreasonable experimental effort' with no guarantee of success.
7. It should be recognized that there are challenges in localizing IRSp53 to the neck of a forming endosome, in addition to the issue of finding the right reagent (a suitable immunoEM antibody against IRSp53). Using the pH pulsing assay we observe IRSp53 recruitment over a larger area which then contracts down to a smaller area just before scission of the vesicle (Figure 5a). By EMs we can only observe the dominant steady state distribution which coincides with the larger patches of membrane labelling that may demarcate the area where CLICs may form. The pH-pulsing gives as a high time resolution, while EM and expansion microscopy gives high spatial resolution - but the combination is what we need to be able to address the point any further than we currently have. It should be noted that dynamin was visualized at the necks of the clathrin-coated vesicle in EM only by using dynamin mutants that arrested endocytosis (Koenig & Ikeda, 1989; Sever, Damke, & Schmid, 2000). Here, finding an arrested clathrin-coated vesicle is relatively easy, since the clathrin coat is electron dense. In the case of CG endocytosis, we do not even have the desired mutants among those tested by us in the manuscript that arrest CG endocytosis a specific stages, nor an electron-dense coat.
8. I-BAR domain of IRSp53 and curvature sensing: We would like to point out that the role of I-BAR domain in curvature sensing is only now beginning to be understood (Prévost et al., 2015). In addition, MD simulations show that although I-BAR crystal structure is

somewhat curved it adopts a flatter structure in solution (Takemura, Hanawa-Suetsugu, Suetsugu, & Kitao, 2017). The paper also suggests that I-BAR does not act as a rigid template. Therefore, we suggest that the I-BAR may bind to the membrane in diverse orientations. Lastly, CG endocytosis requires IRSp53 not just its I-BAR (although a mutation in this domain does not rescue CG endocytosis) and that is regulated by various factors such presence of GTP-CDC42, the activation status of IRSp53, nature of effectors bound to IRSp53 via its SH3 domain.

In summary, we believe that it is well established that IRSp53 binds to the plasma membrane at the site of CG endocytosis. Using pH pulsing assay we have shown it is present at the forming CG endocytic site, until the vesicle forms. This evidence and others presented in the manuscript suggest that IRSp53 is necessary for CG endocytic vesicle formation. But, as the reviewer 1 points out in the absence of unequivocally localizing IRSp53 to the neck we can tone down the claim that the I-BAR domain containing protein is recruited to the neck to help in the endocytic process, and provide the caveats in our discussion.

We hope you find our explanation sufficient and will now find our manuscript suitable for publication in *Nature Communications*.

References cited :

Boucrot, E., Ferreira, A. P. A., Almeida-Souza, L., Debard, S., Vallis, Y., Howard, G., ...

McMahon, H. T. (2015). Endophilin marks and controls a clathrin-independent endocytic pathway. *Nature*, *517*(7535), 460–5. <https://doi.org/10.1038/nature14067>

Koenig, J. H., & Ikeda, K. (1989). Disappearance and reformation of synaptic vesicle membrane upon transmitter release observed under reversible blockage of membrane retrieval. *The Journal of Neuroscience*, *9*(11), 3844–3860.

- Lundmark, R., Doherty, G. J., Howes, M. T., Cortese, K., Vallis, Y., Parton, R. G., & McMahon, H. T. (2008). The GTPase-Activating Protein GRAF1 Regulates the CLIC/GEEC Endocytic Pathway. *Current Biology*, *18*(22), 1802–1808.
<https://doi.org/10.1016/j.cub.2008.10.044>
- Mayor, S., & Pagano, R. E. (2007). Pathways of clathrin-independent endocytosis. *Nat Rev Mol Cell Biol*, *8*(8), 603–612. <https://doi.org/10.1038/nrm2216>
- Mayor, S., Parton, R. G., & Donaldson, J. G. (2014). Clathrin-independent pathways of endocytosis. *Cold Spring Harbor Perspectives in Biology*, *6*(6).
<https://doi.org/10.1101/cshperspect.a016758>
- Prévost, C., Zhao, H., Manzi, J., Lemichez, E., Lappalainen, P., Callan-Jones, A., & Bassereau, P. (2015). IRSp53 senses negative membrane curvature and phase separates along membrane tubules. *Nature Communications*, *6*, 8529.
<https://doi.org/10.1038/ncomms9529>
- Renard, H.-F., Simunovic, M., Lemièrre, J., Boucrot, E., Garcia-Castillo, M. D., Arumugam, S., ... Johannes, L. (2015). Endophilin-A2 functions in membrane scission in clathrin-independent endocytosis. *Nature*, *517*(7535), 493–6.
<https://doi.org/10.1038/nature14064>
- Sever, S., Damke, H., & Schmid, S. L. (2000). Dynamin: GTP controls the formation of constricted coated pits, the rate limiting step in clathrin-mediated endocytosis. *Journal of Cell Biology*, *150*(5), 1137–1147. <https://doi.org/10.1083/jcb.150.5.1137>
- Takemura, K., Hanawa-Suetsugu, K., Suetsugu, S., & Kitao, A. (2017). Salt Bridge Formation between the I-BAR Domain and Lipids Increases Lipid Density and Membrane Curvature. *Scientific Reports*, *7*(1), 6808. <https://doi.org/10.1038/s41598-017-06334-5>